# Improving Federated Learning Face Recognition via Privacy-Agnostic Clusters

**Qiang Meng[1], Feng Zhou[1], Hainan Ren[1], Tianshu Feng[2], Guochao Liu[1], Yuanqing Lin[1]**
[1] Algorithm Research, Aibee Inc. [2] Independent Researcher

## Abstract

The growing public concerns on data privacy in face recognition can be greatly addressed by the federated learning (FL) paradigm. However, conventional FL methods perform poorly due to the uniqueness of the task: broadcasting class centers among clients is crucial for recognition performances but leads to privacy leakage. To resolve the privacy-utility paradox, this work proposes PrivacyFace, a framework largely improves the federated learning face recognition via communicating auxiliary and privacy-agnostic information among clients. PrivacyFace mainly consists of two components: First, a practical Differentially Private Local Clustering (DPLC) mechanism is proposed to distill sanitized clusters from local class centers. Second, a consensus-aware recognition loss subsequently encourages global consensuses among clients, which ergo results in more discriminative features. The proposed framework is mathematically proved to be differentially private, introducing a lightweight overhead as well as yielding prominent performance boosts (*e.g.*, +9.63% and +10.26% for TAR@FAR=1e-4 on IJB-B and IJB-C respectively). Extensive experiments and ablation studies on a large-scale dataset have demonstrated the efficacy and practicability of our method.

## 1 Introduction

Face recognition technique offers great benefits when used in right context, such as public safety, personal security and convenience. However, misuse of this technique is a concern as it involves unique and irrevocable biometric data. The rapid commercial applications based on face recognition and facial analysis techniques have stimulated a global conversation on AI ethics, and have resulted in various actors from different countries issuing governance initiatives and guidelines. EU's General Data Protection Regulation (GDPR) (Voigt & Von dem Bussche, 2017), California Consumer Privacy Act (CCP) and Illinois Personal Information Protection Act (IPI) enforces data protection "by design and by default" in the development of any new framework. On the nationwide "315 show" of year 2021, the China Central Television (CCTV) called out several well-known brands for illegal face collection without explicit user consent. As researchers, it is also our duty to prevent the leakage of sensitive information contained in public datasets widely used by the research community. Therefore, faces in ImageNet (Deng et al., 2009) were recently all obfuscated (Yang et al., 2021) and a large face dataset called MS-Celeb-1M (Guo et al., 2016) was pulled of the Internet.

In the wake of growing social consensus on data privacy, the field of face recognition calls for a fundamental redesign about model training while preserving privacy. A potential solution is the paradigm called Federated Learning (FL) (McMahan et al., 2017a). Given $C$ clients with local datasets $\{\mathcal{D}^1, \mathcal{D}^2, \cdots, \mathcal{D}^C\}$ as shown in Fig. 1a, FL decentralizes the training process by combining local models fine-tuned on each client's private data and thus hinders privacy breaches. Typical examples of these clients include personal devices containing photo collections of a few family members, or open-world scenarios such as tourist attractions visited by tens of thousands of people. In most circumstances, we can safely assume very few classes would co-exist in two or more clients.

Despite the numerous FL-based applications in various domains (Kairouz et al., 2019) ranging from health to NLP, there are very little progress (Aggarwal et al., 2021; Bai et al., 2021; Liu et al., 2021) in training face recognition models with FL schemes. Unlike other tasks, parameters of the last classifier for a face recognition model are crucial for recognition performance but strongly associated with privacy. These parameters can be regarded as mean embeddings of identities (Wang et al., 2018;

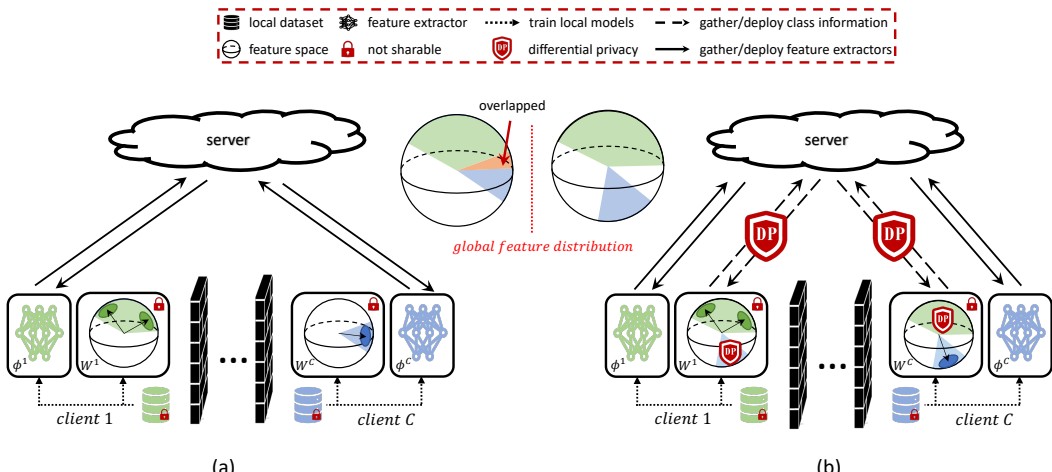

Figure 1: Under the federated setting, multiple clients communicate non-sensitive model parameters $\phi^c$ (excluding the last fully connected layer $\mathbf{W}^c$ which are greatly tied to privacy) under the orchestration by a central server. (a) Since $\mathbf{W}^c$'s are kept locally in conventional FL updating, the embedding space could overlap for different classes during training. (b) In contrast, the proposed PrivacyFace framework learns an improved face embedding by aggregating discriminative embedding clusters that are proved to achieve differential privacy.

Meng et al., 2021b; Shen et al., 2020) (also called as class centers), from where individual privacy could be spied out as studied by plenty of works (Kumar Jindal et al., 2018; Boddeti, 2018; Mai et al., 2020; Dusmanu et al., 2021). That prevents the FL approach from broadcasting the whole model among clients and the central server, and consequently leads to conflicts in the aggregation of local updates. As depicted in the global feature distribution of Fig. 1a, both clients try to spread out their own classes in the same area (pointed by the arrow) of the normalized feature space. Thus, the training loss could oscillate to achieve consensus given the sub-optimal solutions from multiple clients. On the other hand, a large batch with sufficient negative classes is necessary to learn a discriminative embedding space for advanced face recognition algorithms. During the conventional FL updates, each class is only aware of local negative classes while those from other clients are untouchable. This further limits performances of FL approaches in face recognition.

The privacy-utility paradox motivates us to introduce PrivacyFace, a framework improves federated learning face recognition by broadcasting sanitized information of local class globally. In the framework, a novel algorithm called Differentially Private Local Clustering (DPLC) first generates privacy-agnostic clusters of class centers while any specific individual in the cluster cannot be learned, irrespective of attacker's prior knowledge, information source and other holds. Recall that the privacy cost of a differential privacy scheme is propotional to the $l_2$-sensitivity while inversely propotional to the query number. Our DPLC reaches a low $l_2$-sensitivity by restricting the cluster size as well as covering sufficient class centers. In addition, the number of necessary centers to communicate in DPLC is irrelevant to the number of classes in the training data. These characteristics jointly equip DPLC with much smaller privacy cost than the naive alternative, which sanitizes each class center individually by Gaussian noise. In our experiments, DPLC's privacy cost is only 1.7e-7 of that of the naive approach. That persuasively reveals the high security level of our approach.

The second part of PrivacyFace is the consensus-aware face recognition loss. Following principles of Federated Averaging (FedAvg) (McMahan et al., 2017a), a server iteratively gathers feature extractors and privacy-agnostic clusters from clients, averages parameters of feature extractors and distributes them to clients. Accordingly, the consesus-aware loss notifies each client not to embed samples in the inappropriate zone (differential private clusters marked by DP) of the feature space during the local optimization, as shown in Fig. 1b. This process aids each client to train more discriminative features as well as align all consensuses. Compared to the conventional approach, our PrivacyFace boosts performances by +9.63% and +10.26% for TAR@FAR=1e-4 on IJB-B and IJB-C respectively with only single-digit privacy cost. Moreover, the additional computational cost as well as communication cost are negligible (*e.g.*, the extra clusters to broadcast only occupy 16K storage while the backbone already takes 212M). In a word, PrivacyFace is an efficient algorithm which improves conventional federated learning face recognition by a large margin on performances, while requires little privacy cost as well as involves lightweight computational/communication overheads.

## 2 PRELIMINARIES

### 2.1 DEEP FACE RECOGNITION

Most of early works in deep face recognition rely on metric-learning based loss, including contrastive loss (Chopra et al., 2005), triplet loss (Schroff et al., 2015) and N-pair loss (Sohn, 2016). These methods are usually inefficient in training on large-scale datasets. One possible reason is that their embedding spaces at each iteration are constructed only by a positive sample and limited negative ones. Therefore, the main body of research (Ranjan et al., 2017; Liu et al., 2017; Wang et al., 2018; Deng et al., 2019; Xu et al., 2021; Meng et al., 2021a;c) has focused on devising more effective classification-based loss and achieved leading performances on a number of benchmarks. Suppose that we are given a training dataset $\mathcal{D}$ with $N$ face samples $\{x_i, y_i\}_{i=1}^{N}$ of $n$ identities, where each $x_i$ is a face image and $y_i \in \{1, \cdots, n\}$ denotes its associated class label. Considering a feature extractor $\phi$ generating the embedding $\boldsymbol{f}_i = \phi(x_i)$ and a classifier layer with weights $\mathbf{W} = [\boldsymbol{w}_1, \cdots, \boldsymbol{w}_n]$, a series of face recognition losses can be summarized as

$$L_{cls}(\phi, \mathbf{W}) = -\sum_{i=1}^{N} \log \frac{e^{u(\boldsymbol{w}_{y_i}, \boldsymbol{f}_i)}}{e^{u(\boldsymbol{w}_{y_i}, \boldsymbol{f}_i)} + \sum_{j=1, j \neq y_i}^{n} e^{v(\boldsymbol{w}_j, \boldsymbol{f}_i)}}. \tag{1}$$

The specific choices of the similarity functions $u(\boldsymbol{w}, \boldsymbol{f}), v(\boldsymbol{w}, \boldsymbol{f})$ yield different variants, e.g.:

$$\text{CosFace (Wang et al., 2018):} \quad u(\boldsymbol{w}, \boldsymbol{f}) = s \cdot (\cos\theta - m), \quad v(\boldsymbol{w}, \boldsymbol{f}) = s \cdot \cos\theta, \tag{2}$$

$$\text{ArcFace (Deng et al., 2019):} \quad u(\boldsymbol{w}, \boldsymbol{f}) = s \cdot \cos(\theta - m), \quad v(\boldsymbol{w}, \boldsymbol{f}) = s \cdot \cos\theta, \tag{3}$$

where $s, m$ are hyper-parameters and $\theta = \arccos(\boldsymbol{w}^T \boldsymbol{f} / \|\boldsymbol{w}^T \boldsymbol{f}\|)$ is the angle between $\boldsymbol{w}$ and $\boldsymbol{f}$.

### 2.2 DIFFERENTIAL PRIVACY

Differential Privacy (DP) (Dwork et al., 2006) is a well-established framework under which very little about any specific individual can be learned in the process irrespective of the attacker's prior knowledge, information source and other holds (Feldman et al., 2017; Nissim et al., 2016; Stemmer & Kaplan, 2018). We state the relevant definitions and theories (Dwork & Roth, 2014) below.

**Definition 1** (Differential Privacy). *A randomized algorithm $M : \mathcal{X} \to \mathcal{Y}$ is $(\epsilon, \delta)$-DP if for every pair of neighboring datasets $X, X' \in \mathcal{X}$ (i.e., $X$ and $X'$ differ in one row), and every possible output $T \in \mathcal{Y}$ the following equality holds: $Pr[M(X) \in T] \leq e^{\epsilon} Pr[M(X') \in T] + \delta$.*

Here $\epsilon, \delta \geq 0$ are privacy loss parameters, which we consider a privacy guarantee meaningful if $\delta = o(\frac{1}{n})$, where $n$ is the size of dataset.

**Definition 2** ($l_2$-sensitivity). *The $l_2$-sensitivity of a function $f : \mathcal{X} \to \mathbb{R}^d$ is*

$$\Delta_2(f) = \max_{\substack{X, X' \in \mathcal{X} \\ X, X' \text{ are neighbors}}} \|f(X) - f(X')\|_2. \tag{4}$$

**Definition 3** (Gaussian Mechanism). *Suppose a function $f : \mathcal{X} \to \mathbb{R}^d$ and $\mathbf{I}_d$ is the d-dimensional identity matrix. The mechanism $M(X) = f(X) + \mathcal{N}(0, \sigma^2 \mathbf{I}_d)$ is $(\epsilon, \delta)$-DP if $\sigma \geq \frac{\Delta_2(f)}{\epsilon} \sqrt{2 \ln(\frac{1.25}{\delta})}$.*

**Definition 4** (Composition of Differentially Private Algorithms (Dwork & Roth, 2014; Dwork & Lei, 2009)). *Suppose $M = (M_1, M_2, \cdots, M_k)$ is a sequence of algorithms, where $M_i$ is $(\epsilon_i, \delta_i)$-differentially private, and the $M_i$'s are potentially chosen sequentially and adaptively. Then $M$ is $(\sum_{i=1}^{k} \epsilon, \sum_{i=1}^{k} \delta_k)$-differentially private.*

## 3 PRIVACYFACE

This section details the PrivacyFace framework as illustrated in Fig. 2. At its core is a novel clustering algorithm that extracts non-sensitive yet informative knowledge about the local class distribution (Fig. 2a). After the central server broadcasting local DP-guaranteed outputs (Fig. 2b), each client optimizes over a consensus-aware objective that takes into account both the local data and the privacy-agnostic clusters to learn a discriminative embedding space (Fig. 2c) for face recognition.

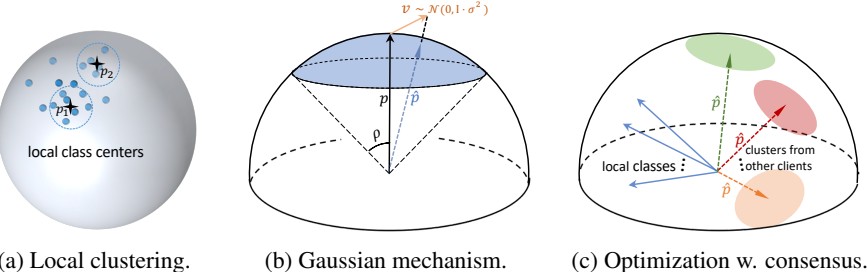

(a) Local clustering.     (b) Gaussian mechanism.     (c) Optimization w. consensus.

Figure 2: Compared to conventional federated learning methods, PrivacyFace learns more discriminative features by three additional steps in each client: (a) Find a cluster with margin $\rho$ and calculate the average $\boldsymbol{p}$ of class centers covered in the cluster; (b) Perturb $\boldsymbol{p}$ with Gaussian noise $\boldsymbol{v}$, which makes outputted $\hat{\boldsymbol{p}}$ to be differentially private. (c) After the server gathering and distributing $\hat{\boldsymbol{p}}$, a consensus-aware face recognition loss enables each class to be separable with local negative classes as well as class clusters from other clients.

## 3.1 Differentially Private Local Clustering

Our target is to improve performances of federated learning face recognition by communicating auxiliary and privacy-agnostic information among clients. To distill useful information from each client and resolve the privacy-utility paradox, we propose a specialized algorithm called Differential Private Local Clustering (DPLC) with rigorous theoretical guarantees.

**Problem Design**. To facilitate the discussion, let us first introduce the margin parameter $\rho \in [0, \pi]$, which defines the boundary of a cluster centered at $\boldsymbol{p}$ by saying that any $\boldsymbol{w}$ in the cluster satisfies[1] $\arccos(\boldsymbol{w}^T \boldsymbol{p}) \leq \rho$. We design the local clustering problem with two principles. First, the clustering results are expected to carry information only about population rather than individual from local dataset. Supported by Theorem 2 proved later, this principle ensures PrivacyFace to gain insight about the underlying distribution, while preserving the privacy of individual record. Second, the spaces confined by the clusters should be tight because they represent the inappropriate zones to escape from in other clients' perspectives. A cluster with large margin $\rho$ would occupy a huge proportion of the sphere, leaving limited room for other embeddings (e.g., half of sphere would be occupied if $\rho = \pi/2$). More specifically, we quantify the occupancy ratio by the following theorem:

**Theorem 1.** *Assume there is a unit and d-dimensional sphere $\mathbb{S}^d$ and an embedding point $\boldsymbol{p} \in \mathbb{S}^d$. Embeddings with angles less than $\rho$ to $\boldsymbol{p}$ (i.e., $\{\boldsymbol{f} : \arccos(\boldsymbol{f}^T \boldsymbol{p}) \leq \rho\}$) occupy $\frac{1}{2} I_{\sin^2(\rho)} \left( \frac{d-1}{2}, \frac{1}{2} \right)$ of the surface area of $\mathbb{S}^d$. Here $I_x(a, b)$ is the regularized incomplete beta function.*

Proved in Sec. A.1.3 of the appendix, Theorem 1 indicates that the occupancy ratio increases monotonically with respect to the margin $\rho$ given a feature dimension $d$. Fig. 3 plots a series of functions between cluster margin $\rho$ and occupancy ratio under different dimensions. Take the rightmost curve corresponding to a typical setting $d = 512$ in face recognition as an example. The occupancy ratios are $0.055, 5 \cdot 10^{-5}, 4 \cdot 10^{-10}$ when $\rho = 1.5, 1.4, 1.3$ respectively. In another word, the whole feature space would be fully occupied if we sample $20,000$ clusters over the original face classes when $\rho = 1.4$. It is obvious

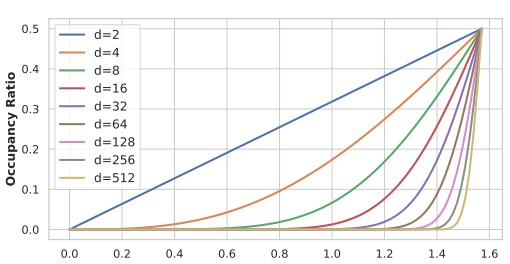

Figure 3: Visualization of the occupancy ratio curves.

that the more space left for optimization, the higher likelihood to achieve better performance. Therefore, we seek for a margin-constrained clustering of local data with fixed $\rho \leq 1.4$ when $d = 512$ for better privacy-utility trade-offs. A counter example to the latter principle is the classical $k$-means algorithm, which have been extended to differentially private versions in a number of works (Nissim et al., 2016; Feldman et al., 2017; Stemmer & Kaplan, 2018). However, $k$-means naturally has no control about the scopes of the generated clusters, not to mention the difficult hyper-parameter $k$ to pick. That motivates us to propose a specified clustering algorithm for our task.

---

[1] We adopt cosine similarity by assuming all vectors lie on a unit sphere in face recognition.

**An Approximated Solution**. Given the margin $\rho$, the problem of finding clusters to cover all class centers is a generalization of the well-studied 2D disk partial covering problem which is unfortunately NP-complete (Xiao et al., 2004). We take a greedy approach by iteratively finding a cluster centered at $\boldsymbol{p}^*$ with margin $\rho$ to cover the most class centers $\boldsymbol{w}_i$ and weeding out those covered before the next round. We cast the problem of identifying such a cluster at each iteration as:

**Definition 5** (Spherical Cap Majority Covering Problem). *Denote $\mathbf{1}(\cdot)$ as an indicator function. Assuming a unit d-dimensional sphere $\mathbb{S}^d$ and class centers $\{\boldsymbol{w}_1, \boldsymbol{w}_2, \cdots, \boldsymbol{w}_n\} \in \mathbb{S}^d$, the target is to find a cluster center $\boldsymbol{p}^* \in \mathbb{S}^d$ where $\boldsymbol{p}^* = \arg\max_{\boldsymbol{p} \in \mathbb{S}^d} \sum_{i=1}^{n} \mathbf{1}(\arccos(\boldsymbol{w}_i^T \boldsymbol{p}) \leq \rho).$*

Finding the optimal $\boldsymbol{p}^*$ is also NP-complete. To address this, we first sort out the densest area $\mathcal{S}$ in terms of the neighbor count for each $\boldsymbol{w}$. An efficient approximation of $\boldsymbol{p}^*$ is then the average of class centers in the area, $\boldsymbol{p} = \frac{1}{|\mathcal{S}|} \sum_{i \in \mathcal{S}} \boldsymbol{w}_i$. Lines 3-5 of Algorithm 1 summarize the detailed procedure, by which the approximate cluster centers $\boldsymbol{p}_i$ can be quickly enumerated as depicted in Fig. 2a. In line 8, the class centers would be further normalized in accordance with the DP mechanism.

---

**Algorithm 1**: Differentially Private Local Clustering (DPLC)

**Data**: Class center embeddings $\mathbf{W} = \{\boldsymbol{w}_1, \boldsymbol{w}_2, \cdots, \boldsymbol{w}_n\}$ encoded in the classifier layer.
**Parameters**: Margin $\rho$; Minimum cluster size $T$; Maximum #queries $Q$; Privacy budget $\epsilon, \delta$.
**Result**: Privacy-agnostic cluster centers $\hat{\boldsymbol{p}}$'s.

1 Let $\mathcal{I} = \{1, 2, \cdots, n\}$ and calculate $\theta_{i,j} = \arccos(\boldsymbol{w}_i^T \boldsymbol{w}_j)$ for $i, j \in \{1, 2, \cdots, n\}$;
2 **for** $q = 1, 2, \cdots Q$ **do**
3    For each $\{w_i, i \in \mathcal{I}\}$, find the indexes of its neighbors by $\mathcal{S}_i = \{j : \theta_{i,j} \leq \rho, j \in \mathcal{I}\}$;
4    Find a set with the most elements, *i.e.*, $\mathcal{S} = \arg\max |\mathcal{S}_i|$ ;
5    $\boldsymbol{p} \leftarrow \frac{1}{|\mathcal{S}|} \sum_{i \in \mathcal{S}} \boldsymbol{w}_i$ ;        // approximate solution for problem 5
6    **if** $|\mathcal{S}| \geq T$ **then**
7      Sample a $\boldsymbol{v}$ from distribution $\mathcal{N}(0, \sigma^2 \mathbf{I}_d)$ where $\sigma = \frac{2}{|\mathcal{S}|\epsilon} \sqrt{(1 - \cos(2\rho)) \ln(\frac{1.25}{\delta})}$ ;
8      $\hat{\boldsymbol{p}}_q \leftarrow \frac{\boldsymbol{p}+\boldsymbol{v}}{\|\boldsymbol{p}+\boldsymbol{v}\|}, \mathcal{I} \leftarrow \mathcal{I}/\{\arccos(\boldsymbol{w}_i^T \frac{\boldsymbol{p}}{\|\boldsymbol{p}\|_2}) \leq \rho, i \in \mathcal{I}\}$ ;
9    **else**
10      Break out of the loop ;

---

**Differential Privacy Endorsement**. Although the centers $\boldsymbol{p}$ reveal the population-level property of the local training set, they are still outcomes of a deterministic algorithm, thereby vulnerable to adversary attack during the FL updating. We prove below that $\boldsymbol{p}$ can be perturbed to achieve DP:

**Theorem 2.** *Define a function $\boldsymbol{p} \triangleq f(\mathcal{S}) = \frac{1}{|\mathcal{S}|} \sum_{i \in \mathcal{S}} \boldsymbol{w}_i$. Then the Gaussian Mechanism $\hat{\boldsymbol{p}} \triangleq M(\mathcal{S}) = f(\mathcal{S}) + \mathcal{N}(0, \sigma^2 \mathbf{I}_d)$ is $(\epsilon, \delta)$-DP if $\sigma \geq \frac{2}{|\mathcal{S}|\cdot\epsilon} \sqrt{(1 - \cos(2\rho)) \ln(\frac{1.25}{\delta})}$.*

*Proof.* Let $\mathcal{S}$ be the set of indexes of class centers which have cosine similarities larger than $\cos\rho$ with respect to a center $\boldsymbol{w}_o$. Assuming that $\mathcal{S}, \mathcal{S}'$ are neighbors differed at $\boldsymbol{w}$ and $\boldsymbol{w}'$, where vectors are normalized, i.e., $\|\boldsymbol{w}\| = \|\boldsymbol{w}'\| = \|\boldsymbol{w}_o\| = 1$, then we have $\boldsymbol{w}_o^T \boldsymbol{w} \geq \cos\rho$ and $\boldsymbol{w}_o^T \boldsymbol{w}' \geq \cos\rho$.

Lemma 2 in appendix states that $\frac{\boldsymbol{w}^T \boldsymbol{w}'}{\|\boldsymbol{w}\|_2 \|\boldsymbol{w}'\|_2} \geq \cos(\arccos \frac{\boldsymbol{w}_o^T \boldsymbol{w}}{\|\boldsymbol{w}_o\|_2 \|\boldsymbol{w}\|_2} + \arccos \frac{\boldsymbol{w}_o^T \boldsymbol{w}'}{\|\boldsymbol{w}_o\|_2 \|\boldsymbol{w}'\|_2})$ for all $\{\boldsymbol{w}, \boldsymbol{w}', \boldsymbol{w}_o\}$. Therefore, the lower bound of $\boldsymbol{w}^T \boldsymbol{w}'$ is

$$\boldsymbol{w}^T \boldsymbol{w}' = \frac{\boldsymbol{w}^T \boldsymbol{w}'}{\|\boldsymbol{w}\|_2 \|\boldsymbol{w}'\|_2} \geq \cos(\arccos \frac{\boldsymbol{w}_o^T \boldsymbol{w}}{\|\boldsymbol{w}_o\|_2 \|\boldsymbol{w}\|_2} + \arccos \frac{\boldsymbol{w}_o^T \boldsymbol{w}'}{\|\boldsymbol{w}_o\|_2 \|\boldsymbol{w}'\|_2})$$
$$\geq \cos(\rho + \rho) = \cos(2\rho)$$

Following this inequality, the $l_2$-sensitivity of $f(\mathcal{S})$ is $\frac{1}{|\mathcal{S}|} \sqrt{2 - 2\cos(2\rho)}$ as $\|f(\mathcal{S}) - f(\mathcal{S}')\|_2 = \frac{1}{|\mathcal{S}|}\|\boldsymbol{w} - \boldsymbol{w}'\|_2 = \frac{1}{|\mathcal{S}|}\sqrt{2 - 2\boldsymbol{w}^T \boldsymbol{w}'} \leq \frac{1}{|\mathcal{S}|}\sqrt{2 - 2\cos(2\rho)}$. By Definition 3, we easily conclude that $M(\mathcal{S}) = \frac{1}{|\mathcal{S}|} \sum_{i \in \mathcal{S}} \boldsymbol{w}_i + \mathcal{N}(0, \sigma^2 \mathbf{I}_d)$ is $(\epsilon, \delta)$-DP if $\sigma \geq \frac{2}{|\mathcal{S}|\cdot\epsilon} \sqrt{(1 - \cos(2\rho)) \ln(\frac{1.25}{\delta})}$. $\square$

Additional proofs can be found in Sec. A.1.2 of appendix. Theorem 2 outlines the setting of noise variance $\sigma$ given the privacy budget $\epsilon, \delta$ and the cluster margin $\rho$. Basically, the lower bound of

$\sigma$ is proportional to $\frac{1}{|\mathcal{S}|}$. That implies when the size of cluster $|\mathcal{S}|$ is large enough, adding small noise is sufficient to achieve promising privacy level. We found setting minimum cluster size to $T \triangleq \min |\mathcal{S}| = 512$ yields empirically stable performance. It is worth emphasizing that the number of original classes $n$ does not directly influence the introduced noise $\sigma$. This advantage ensures that DPLC maintains a high utility when dealing with large-scale face recognition problems.

**Full algorithm**. Algorithm 1 presents the full DPLC and we highlight several key properties below. Properties 1, 2 state that DPLC is efficient and differentially private in theory. Besides the efficency and privacy, another major concern of the DP algorithm is the utility. Specifically, as $\hat{p}$ is a combination of the source cluster center $p$ and Gaussian noise $v$, we should prevent the noise $v$ to have overwhelming effects to the output, which is further verified by our Property 3 and Property 4.

**Property 1.** *The complexity of the DPLC algorithm is $O(Q \cdot n^2)$.*

*Proof.* The computational complexity of calculating $\theta_{i,j}$ is $O(n^2)$. In each loop, we also compare the value $\theta_{i,j}, \forall i, j \in \mathcal{S}$ with $\rho$, whose worst time complexity is $O(n^2)$. Because remaining steps are of linear complexity, the total complexity is $O(n^2 + Q \cdot n^2) = O(Q \cdot n^2)$. $\square$

**Property 2.** *DPLC is a $(Q \cdot \epsilon, Q \cdot \delta)$-differentially private algorithm.*

*Proof.* The mechanism $f(\mathcal{S}) + \mathcal{N}(0, \sigma^2 \mathbf{I}_d)$ is $(\epsilon, \delta)$-differentially private according to Lemma 2. As DPLC queries at most $Q$ results, it's easy to conclude that our DPLC algorithm is $(Q \cdot \epsilon, Q \cdot \delta)$-differentially private based on Definition 4. $\square$

**Property 3.** *Denote $\Phi$ as the cumulative distribution function of a standard Gaussian distribution. For vector magnitudes of $p, v$, we have $\|p\|_2 \in (\cos \rho, 1]$ and $P(\|v\|_2 \leq r) \simeq \Phi(\frac{r^2}{\sigma^2 \sqrt{2(d-1)}} - \frac{d-1}{2})$.*

*Proof.* For $p$, the upper bound is 1 as $\|p\|_2 = \|\frac{1}{|\mathcal{S}|} \sum_{i \in \mathcal{S}} w_i\|_2 \leq \frac{1}{|\mathcal{S}|} \sum_{i \in \mathcal{S}} \|w_i\|_2 = 1$. As $\mathcal{S}$ denotes the set of indexes of class centers which has cosine similarities larger than $\cos \rho$ with a center $w_o$, the lower bound of $\|p\|_2$ can be calculated by

$$\|p\|_2 = \|p\|_2 \|w_o\|_2 \geq p^T w_o = (\frac{1}{|\mathcal{S}|} \sum_{i \in \mathcal{S}} w_i)^T w_o = \frac{1}{|\mathcal{S}|} \sum_{i \in \mathcal{S}} w_i^T w_o \geq \frac{1}{|\mathcal{S}|}(1 + (|\mathcal{S}| - 1) \cos \rho) > \cos \rho.$$

Therefore, we have $\|p\|_2 \in (\cos \rho, 1]$.

For $v \sim \mathcal{N}(0, \sigma^2 \mathbf{I}_d)$, let $v = [v_1, v_2, \cdots, v_d]$, we have $\frac{v_i}{\sigma}$ following the standard normal distribution for all $i$. Thus, the sum of square of $\frac{v_i}{\sigma}$ follows a $\chi^2$-distribution with $d - 1$ degrees of freedom, *i.e.*, $\frac{1}{\sigma^2} \|v\|_2^2 = \sum_{i=1}^d (\frac{v_i}{\sigma})^2 \sim \chi_{d-1}^2$. The mean and variance of $\|v\|_2^2$ are $E(\|v\|_2^2) = \sigma^2(d-1)$ and $Var(\|v\|_2^2) = 2\sigma^4(d-1)$, based on properties of the $\chi^2$-distribution. By the central limit theorem, $\|v\|_2^2$ converges to $\mathcal{N}(\sigma^2(d-1), 2\sigma^4(d-1))$ when $d$ is large enough. In practice, for $d \geq 50$, $\|v\|_2^2$ is sufficiently close to $\mathcal{N}(\sigma^2(d-1), 2\sigma^4(d-1))$ (Box et al., 1978). Therefore, we have

$$P(\|v\|_2 \leq r) = P(\frac{\|v\|_2^2 - \sigma^2(d-1)}{\sqrt{2\sigma^4(d-1)}} \leq \frac{r^2 - \sigma^2(d-1)}{\sqrt{2\sigma^4(d-1)}}) \simeq \Phi(\frac{r^2}{\sigma^2 \sqrt{2(d-1)}} - \sqrt{\frac{d-1}{2}}).$$

Note that in our work, $\sigma$ can be a small value with little privacy cost. Thus, with high probability, the norm of $v$ is close to zero. $\square$

**Property 4.** *If $\|p\|_2 \geq \|v\|_2$, the cosine similarity between $p$ and $\hat{p}$ is always greater than $\sqrt{1 - \|v\|_2^2 / \|p\|_2^2}$.*

*Proof.* Denote magnitudes of $p, v$ as $l_p, l_v$ and cosine similarity between $p$ and $v$ as $x$ (*i.e.*, $x = \frac{p \cdot v}{\|p\| \|v\|} \in [-1, 1]$). Let $a = \frac{l_r}{l_p} < 1$ and $s$ be the cosine similarity between $p + v$ and $p$, then $s = \frac{(p+v)^T p}{\|p+v\|_2 \|p\|_2} = \frac{l_p^2 + l_p l_v x}{l_p \sqrt{l_p^2 + l_v^2 + 2l_p l_v x}} = \frac{1 + ax}{\sqrt{1 + a^2 + 2ax}}$. The first order derivative of $s$ with respect to $x$ is $\frac{ds}{dx} = \frac{a^2(a+x)}{(1+a^2+2ax)^{\frac{3}{2}}}$. It's easy to conclude that $s$ takes the smallest value when $x = -a$, which can be reached as $a$ is always smaller than 1. In the end, we have $s \geq \sqrt{1 - l_v^2 / l_p^2}$. $\square$

In DPLC, we reasonably require large local clusters $\mathcal{S}$ at line 6 while preserving privacy in terms of $\epsilon, \delta$. That constraint ensures the synthesized noise $\boldsymbol{v}$ to have a small magnitude (property 3), which thereby results in slight offsets of the sanitized vector $\hat{\boldsymbol{p}}$ to the source $\boldsymbol{p}$ (property 4). With these theoretical guarantees, DPLC can obtain good end-to-end utility with low privacy cost.

## 3.2 Optimization with Global Consensus

PrivacyFace employs a Federated Learning (FL) paradigm for optimization. Suppose there are $C$ clients, each of which $c = 1, \cdots, C$ owns a training dataset $\mathcal{D}^c$ with $N_c$ images from $n_c$ identities. For client $c$, its backbone is parameterized by $\phi^c$ and the last classifier layer encodes class centers in $\mathbf{W^c} = [\boldsymbol{w}_1^c, \cdots, \boldsymbol{w}_{n_c}^c]$. Without loss of generality, we assume that each class center $\boldsymbol{w}^c$ is normalized. Central to PrivacyFace is the DPLC algorithm, which generates for each client, $Q_c$ clusters defined by the centers $\mathcal{P}^c = \{\hat{\boldsymbol{p}}_1^c, \cdots, \hat{\boldsymbol{p}}_{Q_c}^c\}$ with margin $\rho$. As presented in Algorithm 2, the training alternates the following updates in a sufficient number of rounds $M$.

**Client-side**. In line 3 of Algorithm 2, the server broadcasts current feature extractor $\phi_{t-1}$ to each client. Then each client updates its local cluster centers $\mathcal{P}^c$ that can be safely shared to others in line 5. After receiving $\mathcal{P} = \{\mathcal{P}^1, \cdots, \mathcal{P}^C\}$, each client executes one-round optimization over the consensus-aware face recognition loss on local dataset $\mathcal{D}^c$ in line 7:

$$L^c(\boldsymbol{\phi}^c, \boldsymbol{W}^c) = -\sum_{i=1}^{N_c} \log \frac{e^{u(\boldsymbol{w}_{y_i^c}, \boldsymbol{f}_i^c)}}{e^{u(\boldsymbol{w}_{y_i^c}, \boldsymbol{f}_i^c)} + \sum_{j=1, j \neq y_i^c}^{n_c} e^{v(\boldsymbol{w}_j, \boldsymbol{f}_i^c)} + \sum_{k=1, k \neq c}^{C} \sum_{l=1}^{Q_k} e^{\mu(\hat{\boldsymbol{p}}_l^k, \boldsymbol{f}_i^c, \rho)}}, \quad (5)$$

where $\boldsymbol{f}_i^c$ is the feature extracted by $\phi^c$ on $i$-th instance in client $c$. $\mu(\hat{\boldsymbol{p}}, \boldsymbol{f}, \rho) = s \cdot \cos(\max(\theta - \rho, 0))$ computes the similarity between $\boldsymbol{f}$ and the cluster centered at $\hat{\boldsymbol{p}}$ with margin $\rho$. As illustrated in Fig. 2c, PrivacyFace aims to learn a globally consistent embedding that can not only classify local classes, but also achieve consensuses with other clients on the incompatible clusters $\hat{\boldsymbol{p}}$.

**Server-side**. Similar to other FL approaches, a central server orchestrates the training process and receives the contributions of all clients to the new feature extractor at line 8. The well-known FedAvg (McMahan et al., 2017a) is then utilized to compute an average of all local models. Compared to the conventional FL method as described in Fig. 1a, PrivacyFace empirically achieves better convergence thanks to the consensus-aware loss that implicitly takes the data distribution of other clients into account. Built on the client-wise DPLC algorithm, the framework is potentially compatible with other optimizer, e.g., FedSGD (Shokri & Shmatikov, 2015) as shown in Sec. A.2.1 of appendix.

---

**Algorithm 2**: The PrivacyFace Training Scheme.

---

**Data**: $\mathcal{D}^c$s exclusively owned by each of the $C$ clients; A pre-trained extractor $\phi_0$.
**Parameters**: DPLC-related $\rho, T, Q$; Privacy budget $\epsilon, \delta$; Maximum #communications $M$.
**Result**: A general recognition model $\phi_M$.

1   **for** $t = 1, \cdots, M$ **do**
2     **for** each client $c = 1, \cdots, C$ **do**
3       Synchronize the local extractor $\phi_{t-1}^c = \phi_{t-1}$ as the up-to-date one in server;
4       Generate privacy-agnostic $\mathcal{P}^c = \{\hat{\boldsymbol{p}}_1^c, \hat{\boldsymbol{p}}_2^c, \cdots, \hat{\boldsymbol{p}}_{q_c}^c\}$ via Algorithm 1 ;
5     The server gathers $\mathcal{P}^c$ and distributes $\mathcal{P} = \{\mathcal{P}^1, \mathcal{P}^2, \cdots, \mathcal{P}^C\}$ to all clients;
6     **for** each client $c = 1, \cdots, C$ **do**
7       Update local model $\{\phi_{t-1}^c, \boldsymbol{W}_{t-1}^c\}$ to $\{\phi_t^c, \boldsymbol{W}_t^c\}$ by optimizing the loss (Eq. 5) ;
8       Communicate the feature extractor $\phi_t^c$ to the server while keeping $\boldsymbol{W}_t^c$ locally;
9     The server updates the model by $\phi_t = \frac{1}{C} \sum_{c=1}^{C} \phi_t^c$ ;

---

## 4 Experiments

This section together with the appendix describes extensive experiments on challenging benchmarks to illustrate the superiority of PrivacyFace in training face recognition with privacy guarantee.

**Datasets.** CASIA-WebFace (Yi et al., 2014) contains 0.5M images from 10K celebrities and serves as the dataset for pre-training. BUPT-Balancedface (Wang & Deng, 2020), which comprises four

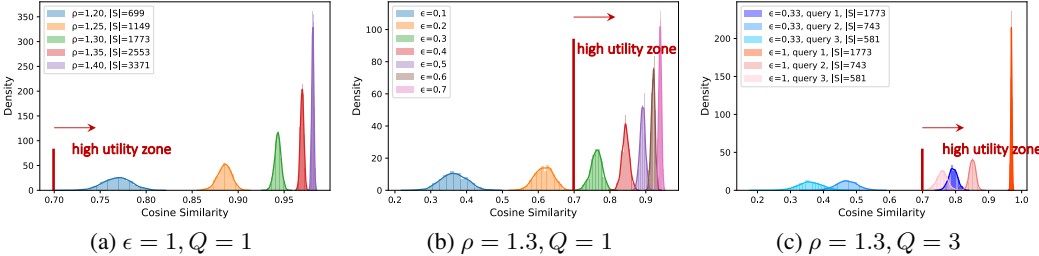

Figure 4: Distributions of cosine similarities of $p$ and $\hat{p}$ under different parameters with 1000 runs.

sub-datasets categorized by racial labels (including African, Asian, Caucasian and Indian) and each sub-dataset contains 7K classes and 0.3M images, is used to simulate the federated setting. We adopt RFW (Wang et al., 2019), IJB-B (Whitelam et al., 2017) and IJB-C (Maze et al., 2018) for evaluation. RFW is proposed to study racial bias and shares the same racial taxonomy as BUPT-Balancedface. IJB-B and IJB-C are challenging ones, containing 1.8K and 3.5K subjects respectively from large-volume in-the-wild images/videos. All images are aligned to $112 \times 112$ based on five landmarks.

**Training.** We assign one client for each of the four sub-datasets and use the perfect federated setting (*i.e.*, no client goes offline during training). For the pre-trained model $\phi_0$, we adopt an open-source one[2] trained on CASIA-WebFace, which builds on ResNet18 and extracts $512$-d features. We finetune $\phi_0$ by SGD for $M = 10$ communication rounds on BUPT-Balancedface, with learning rate $0.001$, batch size $512$ and weight decay $5e\text{-}4$. For reproducibility and fair comparison, all models are trained on 8 1080Ti GPUs with a fixed seed. To alleviate the large domain gaps across sub-datasets, we build a lightweight public dataset with the first 100 classes from CASIA-WebFace to finetune local models before gathered by the server. Unless stated otherwise, the parameters for PrivacyFace are default to $T = 512$, $Q = 1$, $\rho = 1.3$ and $\epsilon = 1$. As the parameter $\delta$ is required to be $O(\frac{1}{|D|})$ (Dwork & Roth, 2014), we set $\delta = \frac{1}{|D|^{1.1}} \approx 5e\text{-}5$ in BUPT-Balancedface.

**Baselines.** In the absence of related methods, we compare PrivacyFace with or without Gaussian noise added, and against a conventional FL method as indicated in Fig. 1a. We denote these methods as $\phi + \hat{p}$, $\phi + p$ and $\phi$ respectively, based on communicated elements among the server and clients. Besides, we implement centralized training on the global version of BUPT-Balancedface and denote the trained model as "global training". The model is also finetuned from $\phi_0$ for 10 epochs with learning rate of 0.001 and serves as an upper bound in all experiments.

**Privacy-Utility Trade-Offs of the DPLC Algorithm.** The goal of DPLC is to broadcast discriminative yet privacy-preserving knowledge through a federated network among clients. This part investigates its effectiveness in terms of similarities between the noise-free center $p$ and the perturbed one $\hat{p}$ by the Gaussian Mechanism with respect to different parameters. We first derive the class centers $W$ from the Caucasian sub-dataset of BUPT-Balancedface by $\phi_0$. Taking $W$ as input, DPLC generates 1000 $\hat{p}'s$ for each $p$, yielding the distribution of cosine similarities shown in Fig. 4. As a common sense, two face features holding cosine similarity over $0.7$ are recognized as from the same identity with a high probability. Fig. 4a reveals that $\rho > 1.2$ can always lead to accurate $\hat{p}$ with $\epsilon = 1$ and $Q = 1$. If $\{\rho, Q\} = \{1.3, 1\}$, we conclude that $\hat{p}, p$ are similar with privacy cost $\epsilon$ over $0.3$, as indicated by Fig. 4b. Although $Q = 1$ leads to good recognition performance, Fig. 4c studies the effect when more queries are required ($Q = 3$) at different cost $\epsilon$. At $\rho = 1.3$, we can generate three clusters with descending sizes (1773, 743, 581) w.r.t the query index. By choosing a strict privacy cost $\epsilon = 0.33$, queries 2 and 3 are too noisy to carry useful information. Setting $\epsilon = 1$ to a reasonable privacy level, however, all queries will convey accurate descriptions of features.

**Ablation Studies on Hyper-parameters.** We conduct several ablation studies on PrivacyFace and present recognition performances on IJB-C in Fig. 5. Models are trained by ArcFace with default parameters if not specifically stated. In Fig. 5a, poor recognition performance and unstable training process can be observed for conventional FL method $\phi$ mainly due to the insufficient number of local classes and inconsistent stationary points achieved by different clients. These drawbacks can be significantly relieved by PrivacyFace $\phi + \hat{p}$. Fig. 5b reveals that performances of PrivacyFace improves as privacy cost $\epsilon$ increases. The performance is nearly saturated when $\epsilon > 0.3$, closing to

---

[2]https://github.com/IrvingMeng/MagFace.

the method $\phi + p$ without privacy protection. This end-to-end evidence indicates DPLC achieves high privacy-utility trade-offs as analyzed in Fig. 4b. The effect of cluster margin $\rho$ is explored in Fig. 5c and the optimal $\rho$ is around 1.3. A small $\rho$ leads to over-fined clusters with inadequate class centers, adding requirements to increase noise and hurt recognition. Alternatively, the performance would drop by adopting a large $\rho$ which generates trivial clusters with high occupancy ratio.

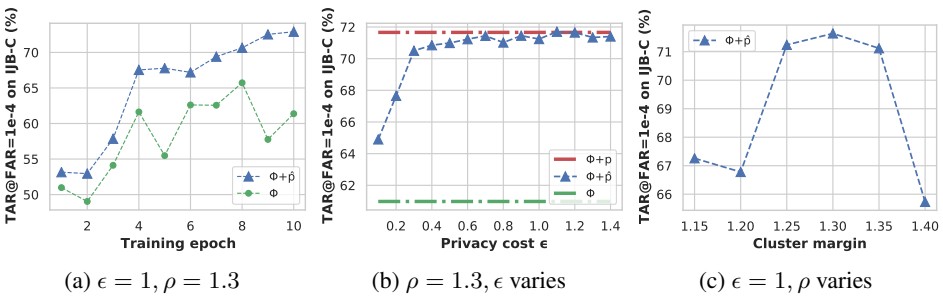

(a) $\epsilon = 1, \rho = 1.3$        (b) $\rho = 1.3, \epsilon$ varies        (c) $\epsilon = 1, \rho$ varies

Figure 5: Effects of hyper-parameters on PrivacyFace performances.

**Performances on Benchmarks.** Tab. 1 presents verification performances on various benchmarks. By finetuning on BUPT-Balancedface, $\phi$, $\phi+\hat{p}$ and $\phi+p$ all achieve performance boosts. Compared to the conventional FL method ($\phi$), performances of our PrivacyFace ($\phi+\hat{p}$) are consistently higher regardless of which training loss used. Specifically, improvements on TAR@FAR=1e-4 on IJB-B and IJB-C are 3.54% and 3.56% with CosFace, and 9.63% and 10.26% with ArcFace, which are significant and demonstrate the superiority of our method. We also observe that $\phi + \hat{p}$ and $\phi + p$ achieve very close results in all benchmarks. That implies the robustness of the proposed method.

Additional experiments can be found in appendix. We implement FedSGD (Shokri & Shmatikov, 2015) in Sec. A.2.1 to show the scalability of the PrivacyFace. Sec. A.2.2 compares the DPLC with a naive approach while Sec. A.2.3 discusses the necessity of FL methods in federated setting. In Sec. A.2.4, we analyze experimental attacks to further verify privacy guarantees of our framework.

Table 1: Verification performances (%) on various benchmarks.

| Loss | Method | RFW | | | | IJB-B | IJB-C |
|---|---|---|---|---|---|---|---|
| | | African | Asian | Caucasian | Indian | TAR@FAR=1e-4 | TAR@FAR=1e-4 |
| - | $\phi_0$ | 81.08 | 82.13 | 89.13 | 86.55 | 5.94 | 8.79 |
| CosFace | $\phi$ | 83.40 | 83.38 | 89.62 | 87.23 | 68.09 | 70.16 |
| | $\phi + \hat{p}$ | 83.50 | 83.38 | 89.93 | 87.28 | 71.63 (**+3.54**) | 73.72 (**+3.56**) |
| | $\phi + p$ | 83.50 | 83.47 | 89.95 | 87.27 | 71.62 | 73.73 |
| | Global Training | 86.30 | 84.58 | 91.48 | 88.92 | 77.35 | 83.20 |
| ArcFace | $\phi$ | 83.50 | 83.08 | 90.26 | 87.32 | 58.62 | 60.98 |
| | $\phi + \hat{p}$ | 83.80 | 83.08 | 90.32 | 87.38 | 68.25 (**+9.63**) | 71.24 (**+10.26**) |
| | $\phi + p$ | 83.82 | 82.97 | 90.32 | 87.38 | 68.57 | 71.66 |
| | Global Training | 87.32 | 84.55 | 92.03 | 88.90 | 71.26 | 79.74 |

**Cost Analysis.** PrivacyFace introduces little computational cost thanks to the efficient DPLC algorithm as well as the consensus-aware loss. Apart from the backbone (over 200M) to distribute as in the conventional FL, the extra variables to communicate are $\hat{p}$'s, which only occupy about 16K storage. Thus, additional communication cost is negligible. Moreover, the total privacy cost is still of a low level with number $M\epsilon = 10$ (*i.e.*, communication rounds times the cost for each round).

# 5 CONCLUSIONS

With the carefully designed DPLC algorithm and a novel consensus-aware recognition loss, we improve federated learning performances on face recognition by communicating auxiliary embedding centers among clients, while achieving rigorous differential privacy. The framework runs efficiently with lightweight communication/computational overheads. Besides, PrivacyFace can be potentially extended to other metric-learning tasks such as re-identification and image retrieval. In the future, more efforts can be spent on designing a more accurate clustering algorithm in conjunction with the FL optimization, *e.g.*, adaptive querying on the confusion areas instead of a brute-force sampling.

## 6    REPRODUCIBILITY STATEMENT

Sec. 3.1 provides key proofs as well as properties of the proposed method. Additional proofs are described in Sec. A.1 of the appendix.

The involved training/test datasets and training configurations are detailed in Sec. 4 when using FedAvg scheme. Similar settings are applied for FedSGD in Sec. A.2.1 in appendix. Sec. A.2.4 of the appendix shows visualizations for two potential attacks: K-nearest neighbor attack and inversion attack. We as well present involved datasets, network structures, training losses as well as training schedules for these attacks. Those are sufficient for reproducibility.

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

## A  APPENDIX

### A.1  MATHEMATICAL PROOFS

In this section, we present extra proofs for PrivacyFace.

#### A.1.1  PROOF FOR THEOREM 1

*Proof.* A spherical cap is a portion of a d-dimensional sphere with radius $r$ cut off by a plane. As shown in (Li, 2011), the surface area of a spherical cap is

$$A = \frac{1}{2} A_d r^{d-1} I_{(2rh-h^2)/r^2} \left( \frac{d-1}{2}, \frac{1}{2} \right).$$

Here $h$ is the height of the cap, and $A, A_d$ are the surface area of the spherical cap and the sphere, respectively. Because the set $\{ \boldsymbol{f} : \arccos(\boldsymbol{f}^T \boldsymbol{p}) \leq \rho, \boldsymbol{f} \in \mathbb{S}^d \}$ is the spherical cap with height $h = r \cdot (1 - \cos \rho)$ and the radius is 1 in our problem, it can be derived that the occupancy ratio is

$$\begin{aligned}
\frac{A}{A_d} &= \frac{1}{2} A_d r^{d-1} I_{(2r^2 \cdot (1-\cos \rho) - r^2 \cdot (1-\cos \rho)^2)/r^2} \left( \frac{d-1}{2}, \frac{1}{2} \right) \\
&= \frac{1}{2} A_d r^{d-1} I_{(2 - 2\cos \rho - 1 - \cos^2 \rho + 2\cos \rho)} \left( \frac{d-1}{2}, \frac{1}{2} \right) \\
&= \frac{1}{2} I_{\sin^2(\rho)} \left( \frac{d-1}{2}, \frac{1}{2} \right).
\end{aligned}$$

$\square$

#### A.1.2  A WEAK VERSION OF THEOREM 2

In this part, we present a straightfward but weak version of the Theorem 2:

**Theorem 3.** *Define a function* $\boldsymbol{p} \triangleq f(\mathcal{S}) = \frac{1}{|\mathcal{S}|} \sum_{i \in \mathcal{S}} \boldsymbol{w}_i$. *Then the Gaussian Mechanism* $\hat{\boldsymbol{p}} \triangleq M(\mathcal{S}) = \frac{1}{|\mathcal{S}|} \sum_{i \in \mathcal{S}} \boldsymbol{w}_i + \mathcal{N}(0, \sigma^2 \mathbf{I}_d)$ *is* $(\epsilon, \delta)$-*DP if* $\sigma \geq \frac{4}{|\mathcal{S}| \cdot \epsilon} \sqrt{(1 - \cos \rho) \ln(\frac{1.25}{\delta})}$.

*Proof.* $\mathcal{S}$ stores indexes of class centers which have cosine similarities larger than $\cos \rho$ with a center (denoted as $\boldsymbol{w}_o$ in this proof). Assuming that $\mathcal{S}, \mathcal{S}'$ are neighbors differed at $w, w'$, then

$$\|f(\mathcal{S}) - f(\mathcal{S}')\|_2 = \frac{1}{|\mathcal{S}|} \|\boldsymbol{w} - \boldsymbol{w}'\|_2 \leq \frac{1}{|\mathcal{S}|} (\|\boldsymbol{w} - \boldsymbol{w}_o\|_2 + \|\boldsymbol{w}' - \boldsymbol{w}_o\|_2) \leq \frac{2}{|\mathcal{S}|} \sqrt{2 - 2\cos \rho} \quad (6)$$

According to Definition 3, we conclude that $M(\mathcal{S}) = \frac{1}{|\mathcal{S}|} \sum_{i \in \mathcal{S}} \boldsymbol{w}_i + \mathcal{N}(0, \sigma^2 \mathbf{I}_d)$ is $(\epsilon, \delta)$-DP if $\sigma \geq \frac{\Delta_2(f)}{\epsilon} \sqrt{2 \ln(\frac{1.25}{\delta})} = \frac{4}{|\mathcal{S}| \cdot \epsilon} \sqrt{(1 - \cos \rho) \ln(\frac{1.25}{\delta})}$. $\square$

This lower bound of $\sigma$ is $\frac{4}{|\mathcal{S}| \cdot \epsilon} \sqrt{(1 - \cos \rho) \ln(\frac{1.25}{\delta})}$, which is larger than our final bound because

$$\begin{aligned}
&(\frac{4}{|\mathcal{S}| \cdot \epsilon} \sqrt{(1 - \cos \rho) \ln(\frac{1.25}{\delta})})^2 - (\frac{2}{|\mathcal{S}| \cdot \epsilon} \sqrt{(1 - \cos(2\rho)) \ln(\frac{1.25}{\delta})})^2 \\
&= \frac{4}{|\mathcal{S}|^2 \cdot \epsilon^2} \ln(\frac{1.25}{\delta}) \cdot (4 - 4\cos \rho - 1 + \cos(2\rho)) \\
&= \frac{4}{|\mathcal{S}|^2 \cdot \epsilon^2} \ln(\frac{1.25}{\delta}) \cdot (3 - 4\cos \rho + 2\cos^2 \rho - 1) \\
&= \frac{4}{|\mathcal{S}|^2 \cdot \epsilon^2} \ln(\frac{1.25}{\delta}) \cdot (2 - 2\cos \rho)^2 \geq 0.
\end{aligned}$$

A tighter bound can reduce the privacy cost with same level of noise added. For example, the privacy cost in our experiments can be dropped by around 21% if using $\rho = 1.3$ (from $\epsilon = 1$ to $\epsilon = 0.793$).

### A.1.3 PROOF FOR THEOREM 1

**Lemma 1.** *For any* $\boldsymbol{x}, \boldsymbol{y}, \boldsymbol{z} \in \mathbb{R}^d$, *the following inequality holds:*

$$(\boldsymbol{x}^T\boldsymbol{x}\boldsymbol{z}^T\boldsymbol{z} - \boldsymbol{x}^T\boldsymbol{z}\boldsymbol{x}^T\boldsymbol{z}) \cdot (\boldsymbol{y}^T\boldsymbol{y}\boldsymbol{z}^T\boldsymbol{z} - \boldsymbol{y}^T\boldsymbol{z}\boldsymbol{y}^T\boldsymbol{z}) \geq (\boldsymbol{x}^T\boldsymbol{z}\boldsymbol{z}^T\boldsymbol{y} - \boldsymbol{x}^T\boldsymbol{y}\boldsymbol{z}^T\boldsymbol{z})^2.$$

*Proof.* Let $\boldsymbol{x} = (x_1, x_2, \cdots, x_d), \boldsymbol{y} = (y_1, y_2, \cdots, y_d), \boldsymbol{z} = (z_1, z_2, \cdots, z_d)$. Then

$$
\begin{aligned}
\boldsymbol{x}^T\boldsymbol{x}\boldsymbol{z}^T\boldsymbol{z} - \boldsymbol{x}^T\boldsymbol{z}\boldsymbol{x}^T\boldsymbol{z} &= \sum_{i=1}^d x_i^2 \cdot \sum_{j=1}^d z_j^2 - (\sum_{i=1}^d x_i z_i)^2 \\
&= \sum_{i=1}^d \sum_{j=1}^d x_i^2 z_j^2 - \sum_{i=1}^d x_i z_i \cdot \sum_{j=1}^d x_j z_j \\
&= \sum_{i=1}^d \sum_{j=1}^d x_i^2 z_j^2 - \sum_{i=1}^d \sum_{j=1}^d x_i z_i x_j z_j \\
&= \sum_{i=1}^d \sum_{j=1}^d (x_i^2 z_j^2 - x_i z_i x_j z_j) \\
&= \frac{1}{2} \sum_{i=1}^d \sum_{j=1}^d (x_i^2 z_j^2 + x_j^2 z_i^2 - 2 x_i z_i x_j z_j) \\
&= \frac{1}{2} \sum_{i=1}^d \sum_{j=1}^d (x_i z_j - x_j z_i)^2
\end{aligned}
$$

Similarly, it's easy to prove $\boldsymbol{y}^T\boldsymbol{y}\boldsymbol{z}^T\boldsymbol{z} - \boldsymbol{y}^T\boldsymbol{z}\boldsymbol{y}^T\boldsymbol{z} = \frac{1}{2}\sum_{i=1}^d \sum_{j=1}^d (y_i z_j - y_j z_i)^2$. In addition, we have

$$
\begin{aligned}
\boldsymbol{x}^T\boldsymbol{z}\boldsymbol{z}^T\boldsymbol{y} - \boldsymbol{x}^T\boldsymbol{y}\boldsymbol{z}^T\boldsymbol{z} &= \sum_{i=1}^d x_i z_i \cdot \sum_{j=1}^d z_j y_j - \sum_{i=1}^d x_i y_i \cdot \sum_{j=1}^d z_j z_j \\
&= \sum_{i=1}^d \sum_{j=1}^d (x_i z_i z_j y_j - x_i y_i z_j z_j) \\
&= \frac{1}{2} \sum_{i=1}^d \sum_{j=1}^d (x_i z_i z_j y_j - x_i y_i z_j z_j + x_j z_j z_i y_i - x_j y_j z_i z_i) \\
&= \frac{1}{2} \sum_{i=1}^d \sum_{j=1}^d (x_i z_j \cdot (z_i y_j - y_i z_j) + x_j z_i \cdot (z_j y_i - y_j z_i)) \\
&= \frac{1}{2} \sum_{i=1}^d \sum_{j=1}^d (x_i z_j - x_j z_i)(z_i y_j - y_i z_j)
\end{aligned}
$$

Then we can deduce that

$$
\begin{aligned}
(\boldsymbol{x}^T\boldsymbol{x}\boldsymbol{z}^T\boldsymbol{z} - \boldsymbol{x}^T\boldsymbol{z}\boldsymbol{x}^T\boldsymbol{z}) \cdot (\boldsymbol{y}^T\boldsymbol{y}\boldsymbol{z}^T\boldsymbol{z} - \boldsymbol{y}^T\boldsymbol{z}\boldsymbol{y}^T\boldsymbol{z}) &= \frac{1}{2}\sum_{i=1}^d \sum_{j=1}^d (x_i z_j - x_j z_i)^2 \frac{1}{2}\sum_{i=1}^d \sum_{j=1}^d (y_i z_j - y_j z_i)^2 \\
&\geq \frac{1}{4}\left(\sum_{i=1}^d \sum_{j=1}^d (x_i z_j - x_j z_i)(z_i y_j - y_i z_j)\right)^2 \\
&= (\boldsymbol{x}^T\boldsymbol{z}\boldsymbol{z}^T\boldsymbol{y} - \boldsymbol{x}^T\boldsymbol{y}\boldsymbol{z}^T\boldsymbol{z})^2
\end{aligned}
$$

Here the second step is the Cauchy–Schwarz inequality. $\qquad\square$

**Lemma 2.** *Assume that* $\boldsymbol{w}, \boldsymbol{w}', \boldsymbol{w}_o \in \mathbb{R}^d$ *and* $\frac{\boldsymbol{w}_o^T\boldsymbol{w}}{\|\boldsymbol{w}_o\|_2\|\boldsymbol{w}\|_2} = \cos\alpha, \frac{\boldsymbol{w}_o^T\boldsymbol{w}'}{\|\boldsymbol{w}_o\|_2\|\boldsymbol{w}'\|_2} = \cos\beta, 0 \leq \alpha, \beta \leq \frac{\pi}{2}$. *Then* $\frac{\boldsymbol{w}^T\boldsymbol{w}'}{\|\boldsymbol{w}\|_2\|\boldsymbol{w}'\|_2} \geq \cos(\alpha + \beta)$ *always holds.*

*Proof.* Let $\frac{\boldsymbol{w}^T \boldsymbol{w}'}{\|\boldsymbol{w}\|_2 \|\boldsymbol{w}'\|_2} = \cos\gamma$. We thereby need to prove $\cos\gamma \geq \cos(\alpha + \beta) = \cos\alpha\cos\beta - \sin\alpha\sin\beta = \cos\alpha\cos\beta - \sqrt{(1 - \cos^2\alpha)(1 - \cos^2\beta)}$, which is equivalent to

$$\sqrt{(1 - \cos^2\alpha)(1 - \cos^2\beta)} \geq \cos\alpha\cos\beta - \cos\gamma. \tag{7}$$

If $\cos\alpha\cos\beta \leq \cos\gamma$, the inequality equation 7 holds.

If $\cos\alpha\cos\beta > \cos\gamma$, then

$$(1 - \cos^2\alpha)(1 - \cos^2\beta) \geq (\cos\alpha\cos\beta - \cos\gamma)^2$$
$$\Longleftrightarrow (1 - (\frac{\boldsymbol{w}_o^T \boldsymbol{w}}{\|\boldsymbol{w}_o\|_2 \|\boldsymbol{w}\|_2})^2)(1 - (\frac{\boldsymbol{w}_o^T \boldsymbol{w}'}{\|\boldsymbol{w}_o\|_2 \|\boldsymbol{w}'\|_2})^2) \geq (\frac{\boldsymbol{w}_o^T \boldsymbol{w}}{\|\boldsymbol{w}_o\|_2 \|\boldsymbol{w}\|_2} \frac{\boldsymbol{w}_o^T \boldsymbol{w}'}{\|\boldsymbol{w}_o\|_2 \|\boldsymbol{w}'\|_2} - \frac{\boldsymbol{w}^T \boldsymbol{w}'}{\|\boldsymbol{w}\|_2 \|\boldsymbol{w}'\|_2})^2$$
$$\Longleftrightarrow ((\|\boldsymbol{w}_o\|_2 \|\boldsymbol{w}\|_2)^2 - (\boldsymbol{w}_o^T \boldsymbol{w})^2)(\|\boldsymbol{w}_o\|_2 \|\boldsymbol{w}'\|_2)^2 - (\boldsymbol{w}_o^T \boldsymbol{w}')^2) \geq ((\boldsymbol{w}_o^T \boldsymbol{w}) \cdot (\boldsymbol{w}_o^T \boldsymbol{w}') - \boldsymbol{w}^T \boldsymbol{w}' \|\boldsymbol{w}_o\|_2^2)^2$$
$$\Longleftrightarrow (\boldsymbol{w}^T \boldsymbol{w} \boldsymbol{w}_o^T \boldsymbol{w}_o - \boldsymbol{w}^T \boldsymbol{w}_o \boldsymbol{w}^T \boldsymbol{w}_o) \cdot (\boldsymbol{w}'^T \boldsymbol{w}' \boldsymbol{w}_o^T \boldsymbol{w}_o - \boldsymbol{w}'^T \boldsymbol{w}_o \boldsymbol{w}'^T \boldsymbol{w}_o) \geq (\boldsymbol{w}^T \boldsymbol{w}_o \boldsymbol{w}_o^T \boldsymbol{w}' - \boldsymbol{w}^T \boldsymbol{w}' \boldsymbol{w}_o^T \boldsymbol{w}_o)^2.$$

The last step is correct according to Lemma 1. □

## A.2 EXTRA EXPERIMENTS

### A.2.1 PRIVACYFACE WITH FEDSGD

Algorithm 2 is mainly built on FedAvg (McMahan et al., 2017a) where model parameters are communicated between clients and the server. In this part, we show that our PrivacyFace is also compatible with another popular FL method called FedSGD (Shokri & Shmatikov, 2015). FedSGD mainly differs FedAvg in two aspects: (1) During training locally, the model is frozen to ensure gradients across mini-batches are from a same model. (2) Gradients rather than model parameters are aggregated and distributed among clients and the server. Apart from changing FedAvg to FedSGD, we let other training setting consistent with those described before and present the results in Tab. 2. Compared to the conventional FL method $\phi$, performances of our PrivacyFace $\phi + \hat{\boldsymbol{p}}$ are close to those on RFW and noteworthily higher in IJB-B/IJB-C benchmarks. Besides, $\phi + \hat{\boldsymbol{p}}$ and $\phi + \boldsymbol{p}$ achieve very close results. All these phenomenons are in accord with those observed when using FedAvg, which shows the scalability of the proposed PrivacyFace.

However, FedSGD does not achieve that significant improvements as FedAvg. A possible reason is that FedSGD naturally requires more communication rounds than FedAvg to achieve comparable results, which may be caused by low-frequent updates during training. Experiments in McMahan et al. (2017a) showed that FedSGD requires about 23 times more communication rounds to achieve similar performances with FedAvg. With such a fact, the improvements using FedSGD in PrivacyFace is still remarkable.

Table 2: Verification performances (%) on various benchmarks with FedSGD and ArcFace.

| Method | RFW | | | | IJB-B | IJB-C |
|--------|-----|-----|-----|-----|-------|-------|
| | African | Asian | Caucasian | Indian | TAR@FAR=1e-4 | TAR@FAR=1e-4 |
| $\phi_0$ | 81.08 | 82.13 | 89.13 | 86.55 | 5.94 | 8.79 |
| $\phi$ | 81.40 | 82.38 | 89.08 | 86.68 | 63.24 | 68.35 |
| $\phi + \hat{\boldsymbol{p}}$ | 81.40 | 82.35 | 89.12 | 86.77 | 63.72(+0.49) | 68.73(+0.38) |
| $\phi + \boldsymbol{p}$ | 81.40 | 82.35 | 89.12 | 86.77 | 63.72 | 68.73 |
| Global Training | 87.32 | 84.55 | 92.03 | 88.90 | 71.26 | 79.74 |

### A.2.2 COMPARISON WITH A NAIVE DP APPROACH

A naive alternative to DPLC is to directly make each class center of $\boldsymbol{W}$ to be differential private by adding noise. However, that can lead to a large privacy cost due to two reasons:

1. With the same noise added, the privacy cost is propotional to the $l_2$-sensitivity based on definition 2. The $l_2$-sensitivity of the naive approach is 2 as maximum distances between two class centers distributed in a unit sphere are 2. In contrast, the $l_2$-sensitivity DPLC is $\frac{1}{|\mathcal{S}|}\sqrt{2 - 2\cos(2\rho)}$. In our experiment, we let $|\mathcal{S}| > 512$ and $\rho = 1.4$, which makes the $l_2$-sensitivity to be smaller than 0.0024.

2. The training dataset are of a large number of identities. According to the composition rule (Definition 3), the privacy cost grows linearly with respect to the queries (*i.e.*, number of identities). In DPLC, low privacy cost as the clustering mechanism highly reduce the number of queries.

During training, the number of query is 1. Therefore, the privacy cost in an local client (with 7000 classes) of DPLC is $\frac{1 \cdot 0.024}{7000 \cdot 2} \approx 1.7e - 7$ of that of the naive approach.

We experimentally compare the performances of our method and the naive approach. For fair comparisons, the privacy cost in each round is fixed to be 1 and all training settings are consistent with those in the main text. Tab. 3 are the results, where the $\phi + \hat{W}$ represents the naive approach. It can be seen that $\phi + \hat{W}$ can achieve similar results on RFW benchmark as learning local discriminative features with introduced fixed class centers are relatively easy (for example, if introduced class center distributed in the south of the embedding space, placing local features to the north is a solution). However, as the class centers from other clients are very noisy, the global embedding space will become a mess. $\phi + \hat{W}$ even achieves much worse results than the conventional FL methods $\phi$ on the challenging benchmarks. The tar decreases 23.99% on IJB-B and 25.16% on IJB-C when far is 1e-4. In contrast, PrivacyFace increases the performances by 9.63% and 10.26% respectively, which shows the superiority of our framework.

Table 3: Verification performances (%) on various benchmarks.

| Method | RFW | | | | IJB-B | IJB-C |
|---|---|---|---|---|---|---|
| | African | Asian | Caucasian | Indian | TAR@FAR=1e-4 | TAR@FAR=1e-4 |
| $\phi$ | 83.50 | 83.08 | 90.26 | 87.32 | 58.62 | 60.98 |
| $\phi + \hat{W}$ | 84.03 | 83.58 | 90.33 | 87.26 | 34.63 (-23.99) | 35.82 (-25.16) |
| $\phi + \hat{p}$ | 83.80 | 83.08 | 90.32 | 87.38 | 68.25 (+9.63) | 71.24 (+10.26) |

### A.2.3 NECESSITY OF FEDERATED LEARNING

Besides using federated learning, one can also choose to finetune client-expert models on local datasets. We also implement this approach by training four independent models on sub-datasets of BUPT-Balancedface and test their performances on benchmarks. The training procedure is consistent with what we described before in Sec. 4. Specifically, we finetune each model from $\phi_0$ by ArcFace for 10 epochs with learning rate 0.001, batch size 512 and weight decay 5e-4.

We present our results in Tab. 4, where the "African/Asian/Caucasian/Indian" in the first column means the model trained on the corresponding sub-dataset of BUPT-Balancedface. Models finetuned on African and Caucasian sub-dataset achieve satisfying results on local benchmarks while those on Asian and Indian have even worse performances than the initial model $\phi_0$. Moreover, all these local models perform poorly on the IJB-B and IJB-C benchmarks, which indicates the lack of generality. To conclude, local training highly relies on the properties and qualities of local datasets, and can not lead to general models. In contrast, models trained by conventional FL method and PrivacyFace can achieve competitive performances on all benchmarks, and therefore are more robust to various scenarios.

Table 4: Verification performances (%) on various benchmarks.

| Method | RFW | | | | IJB-B | IJB-C |
|---|---|---|---|---|---|---|
| | African | Asian | Caucasian | Indian | TAR@FAR=1e-4 | TAR@FAR=1e-4 |
| $\phi_0$ | 81.08 | 82.13 | 89.13 | 86.55 | 5.94 | 8.79 |
| African | 87.53 | - | - | - | 8.56 | 11.26 |
| Asian | - | 64.08 | - | - | 0.30 | 0.40 |
| Caucasian | - | - | 91.87 | - | 53.51 | 57.89 |
| Indian | - | - | - | 83.62 | 39.01 | 39.62 |
| Conventional FL | 83.50 | 83.08 | 90.26 | 87.32 | 58.62 | 60.98 |
| PrivacyFace | 83.80 | 83.08 | 90.32 | 87.38 | 68.25 | 71.24 |

### A.2.4 VISUALIZATION

In this section, we experimentally examine the privacy attacks to demonstrate that class centers lead to privacy leakage while our sanitized clusters are resistent to these attacks. We mainly consider two types of attacks: K-nearest neighbor attack and reversion attack.

**K-Nearest Neighbor Attack** We assume that an attacker owns a gallery of faces with enormous identities and attacks the exposed class centers by finding their neighbors. To simulate the K-nearest neighbor attack, class centers are extracted on the BUPT-Balancedface dataset and three faces are sampled from each identity on a large dataset called MS1M-V2 (Deng et al., 2019). Fig. 6 presents the results. It can be observed that the K-nearest neighbor attack successfully find faces of the corresponding identity from the class centers, which enables the attacker to know which identities are contained in the training dataset. The observation further verifies the fact that sharing class centers leads to privacy leakage in face recognition.

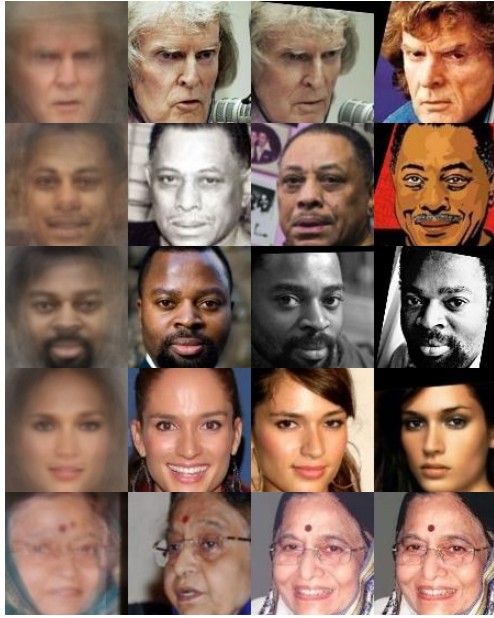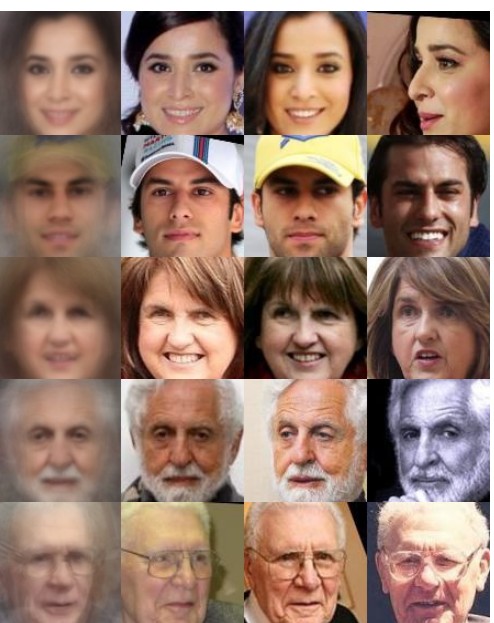

Figure 6: Real mean faces and the top-3 nearest faces of the corresponding mean class centers.

**Reversion Attack** We design a reconstruction network (modified from DCGAN (Radford et al., 2015)) as described in Tab. 5 which projects a 512 dimensional embedding to a face image. CASIA-WebFace dataset is used as the training dataset. Specifically, we extract features from images by the initial model $\phi_0$. With supervision of features and images, we train the network by Adam optimizer with L1 loss, which penalizes the absolute value of pixel differences between reconstructed and real images. The learning rate is initialized from $0.001$ and divided by 10 every 10 epochs, and we stop the training at the 50th epoch. The weight decay is $5e-4$ and batch size is $512$ during our training.

Table 5: Details of the reconstruction network. Note that all the convtranspose2d layers use stride 2 and kernel size $4 \times 4$.

| Layer type | input dim. | output dim. |
|---|---|---|
| fc + BN | 512 | 8192 |
| reshape | 8192 | [512, 4, 4] |
| convtranspose2d + BN + ReLU | [512, 4, 4] | [256, 8, 8] |
| convtranspose2d + BN + ReLU | [256, 8, 8] | [128, 16, 16] |
| convtranspose2d + BN + ReLU | [128, 16, 16] | [64, 32, 32] |
| convtranspose2d + BN + ReLU | [64, 32, 32] | [32, 64, 64] |
| convtranspose2d + Sigmoid | [32, 64, 64] | [3, 128, 128] |

We test our model on BUPT-Balancedface dataset and present reconstruction visualizations. In each image pair of Fig. 7, the left one is the average of faces from one identity while the right one is the reconstructed image from the corresponding class center. It can be seen that reconstructed images reveal the real individual identity, which demonstrates that $w$ carries human face privacy and therefore should be protected. Fig. 8 shows the reconstructions from $p$ and $\hat{p}$ under various privacy cost $\epsilon$. The left column is the images by $p$, which are much blurred than those from $w$ and only the group properties can be identified. For example, the first image seems like an African male while

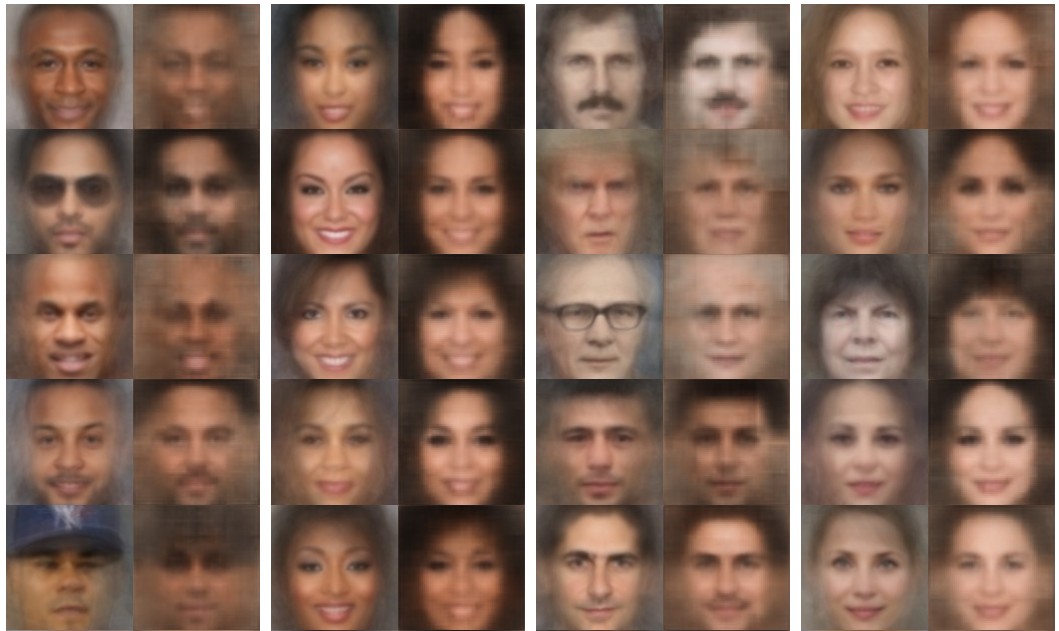

Figure 7: Real mean faces (first image in each pair) and faces reconstructed from class centers $w$ (second image in each pair).

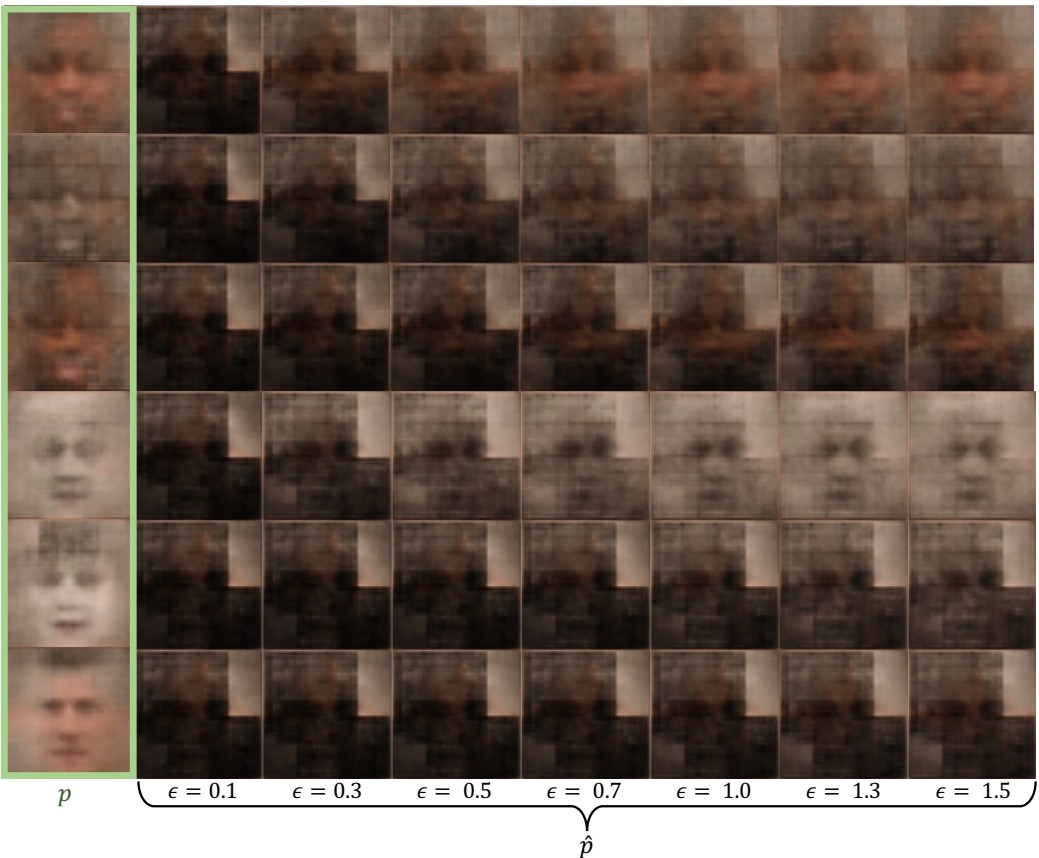

Figure 8: Reconstructed faces from cluster centers $p$ and the differentially private ones $\hat{p}$ under different privacy cost $\epsilon$.

the second last one are very likely to be Caucasian female. The output $\hat{p}$ from our DPLC algorithm

can further protect privacy as one can hardly recognize a face from the generated images, especially when $\epsilon$ is small. The hardness of a successful identification is negative related to $\epsilon$ according to the visualization.

### A.2.5 COMPARISONS WITH FEDFACE

FedFace (Aggarwal et al., 2021) is a recent work applying federated learning to face recognition. In each communication round, the FedFace uploads all local class centers to the server and consequently utilizes the spreadout regularizer to separate class centers from different clients. In this section, we adjust FedFace to fit our federated setting in the main text. Specifically, in each communication round, each local client uploads its 7K class centers to the server. The spreadout regularizer then computes distances of 588 million ($7K \times 7K \times 3 \times 4$) pairs of class centers and back propagates the model parameters once. The updated parameters are then distributed to each of the clients. However, directly computing all distances consumes 11,962M GPU memory, which is out of the capacity of our machine. Therefore, we sample 1K out of 7K class centers from each client for spreadout regularizer in our machine. For hyper-parameters of the spreadout regularizer, the authors recommend loss weight $\lambda = 10$ and margin $v = 0.9$. For fair comparisons, we further run 18 experiments with $\lambda \in \{0.5, 1, 1.5, 5, 10, 15\}$ and $v \in \{0.3, 0.6, 0.9\}$ and report the best results of FedFace.

Table 6: Comparisons of FedFace and PrivacyFace.

| Method | TAR@FAR=1e-4 (%) | | Privacy | Additional cost per round | | |
|---|---|---|---|---|---|---|
| | IJB-B | IJB-C | Cost | Communication | Computation | GPU memory |
| FedAvg | 58.62 | 60.98 | 0 | 0 | 0 | 0 |
| FedFace | 59.40 | 63.25 | $\infty$ | 14,000 vectors | High | 1525M |
| PrivacyFace | 68.25 | 71.24 | 1 | 5 vectors | Low | < 1M |

Tab. 6 summarizes the overall comparisons between FedFace and PrivacyFace. We observe that

- **Performance**. FedFace only boosts the FedAvg by $0.82\%, 2.27\%$ on TAR@FAR=1e-4 for IJB-B and IJB-C. In contrast, PrivacyFace achieves $9.63\%, 10.26\%$, respectively. Besides the sampling strategy, the main reason can be that FedFace only separates classes from different clients for one time (the spreadout regularizer) in a communication round, while the model is locally updated 0.3 million times (number of local images) without considering the conflicts among clients. In contrast, our consensus-aware recogntion loss takes the conflicts into account for each local update.

- **Privacy cost**. In each round, FedFace uploads $7,000$ original class centers to the server. Therefore, the privacy cost is $7000 \times \infty = \infty$. For PrivacyFace, the number is only 1 from the outputted noisy cluster center in the client.

- **Additional communication cost**. For each client in communication round, FedFace additionally uploads $7,000$ 512-d class centers to the server and sends back the updated class centers. In PrivacyFace, a local client uploads one 512-d cluster centers to the server and downloads 4 cluster centers from the server. The additional cost from FedFace is $\frac{7000+7000}{1+4} = 2,800$ times of that from PrivacyFace.

- **Additional computational cost**. The additional computations in FedFace is from the spreadout regularizer, which calculates 588 million pairs of 512-d features and updates the model parameters on the server side by one-time back propagation. That requires enormous computation in each round. In contrast, only extra items in the consensus-aware loss requires additional computation in PrivacyFace, which is of tiny volume.

- **Additional usage of GPU memory**. Even though we sample $1,000$ out of $7,000$ clients to reduce GPU usage of the spreadout regularizer, FedFace still consumes $1,525M$ extra GPU memories. One the other hand, the additional usage of PrivacyFace is negligible, as shown in Tab. 6.

### A.2.6 ABLATION STUDIES ON INITIAL MODELS

In this section, we examine the effects of different initial models on the proposed method. Besides the $\phi_0$ we used before, we further adopt two intermediate checkpoints at epoch 20, 10 when training on the CASIA-WebFace (Yi et al., 2014) and denote them as $\phi_1, \phi_2$, respectively. One additional

model $\phi_3$ is trained on MS1M-V2 (Deng et al., 2019) for 10 epochs with learning rate 0.1 and the ArcFace loss. Note that we do not follow the conventional training configuration where the learning rate is decreased stepwisely until 0.001. That's because such a configuration will lead to a strong model that does not require finetuning on BUPT-BalancedFace (Wang & Deng, 2020).

With other training settings same as those in the main text, we finetune the models on BUPT-BalancedFace (Wang & Deng, 2020) with the ArcFace loss. Tab. 7 presents the comparisons of plain FedAvg and PrivacyFace. On the IJB-B/IJB-C benchmarks, PrivacyFace can consistently achieve better results than the plain FedAvg. For TAR@FAR=1e-4 on IJB-C, PrivacyFace can boost the performances by $10.26\%, 12.03\%, 3.43\%, 22.00\%$ with the four initial models. It can be observed that improvements are affected by the initial model. For example, the poorest initial model $\phi_2$ achieves the lowest improvements. The reason may be that the training loss are overwhelmed by separating local classes as initial features are not discriminative enough. Consequently, increasing inter-client distances is of the secondary importance to the training in this scenario. For the best initial model $\phi_3$, the performances on IJB-B/C decreases when using federated learning. This is because the finetuning process also focuses on improving local performances as the global consensus is already reached. In this senario, FedAvg decreases the TAR@FAR=1e-4 by more than $40\%$ on the benchmarks. In contrast, PrivacyFace can highly alleviate the performance degradations by taking global consensus into account during training. Finetuing on BUPT-BalancedFace mainly benefits the performances on the RFW benchmarks as these two datasets are both designed for racial bias. If the initial model is good enough, the finetuning procedure will focus more on fitting the racial issue instead of improving the general performances. This may be the reason why model $\phi_3$ leads to the largest improvements.

Table 7: Verification performances (%) with different initial models.

| Initial Model | Method | RFW | | | | IJB-B TAR@FAR=1e-4 | IJB-C TAR@FAR=1e-4 |
|---|---|---|---|---|---|---|---|
| | | African | Asian | Caucasian | Indian | | |
| $\phi_0$ | $\phi_0$ | 81.08 | 82.13 | 89.13 | 86.55 | 5.94 | 8.79 |
| | FedAvg | 83.50 | 83.08 | 90.26 | 87.32 | 58.62 | 60.98 |
| | PrivacyFace | 83.80 | 83.08 | 90.32 | 87.38 | 68.25 (**+9.63**) | 71.24 (**+10.26**) |
| $\phi_1$ | $\phi_1$ | 81.37 | 82.13 | 89.22 | 86.75 | 2.04 | 2.56 |
| | FedAvg | 83.33 | 81.70 | 89.73 | 86.65 | 50.70 | 53.97 |
| | PrivacyFace | 83.43 | 82.47 | 90.02 | 86.97 | 61.97 (**+11.27**) | 66.00 (**+12.03**) |
| $\phi_2$ | $\phi_2$ | 76.72 | 77.90 | 84.90 | 82.11 | 1.72 | 1.63 |
| | FedAvg | 78.01 | 77.62 | 85.03 | 82.78 | 35.83 | 36.17 |
| | PrivacyFace | 78.93 | 78.50 | 85.55 | 83.05 | 38.47 (**+2.64**) | 39.6 (**+3.43**) |
| $\phi_3$ | $\phi_3$ | 83.65 | 84.45 | 89.08 | 87.55 | 69.20 | 72.00 |
| | FedAvg | 87.07 | 86.48 | 92.08 | 89.15 | 26.70 | 26.14 |
| | PrivacyFace | 87.60 | 86.80 | 92.37 | 89.85 | 48.10 (**+21.40**) | 48.14 (**+22.00**) |

### A.2.7 EFFECTS OF CLIENT DISCONNECTIONS

Previously we use a perfect federated setting where all clients are available during training. In real-world applications, some clients may drop for certain rounds of communication due to network issues. We further simulate the case by randomly select one of the four clients and exclude it from the training for an offline probability in each communication round. For example, if we set the offline probability to 100%, there will be exactly one client offline per round. Note here we only block one client because the total number of clients is small. We keep other settings unchanged and compare performances of PrivacyFace and plain FedAvg on the IJB-C benchmark with different offline probability. We refer the offline policy as disconnecting patterns during training, and use the same offline policy for each pair of experiments on PrivacyFace and FedAvg for fair purpose. To reduce the randomness, we run each experiment with 3 different offline policies and report the mean TAR@FAR=1e-4.

Fig. 9 presents our results. When the offline probability is 25% and 50%, PrivacyFace improves the performances of federated learning by about 6%. The improvement decreases to 2.4% with 75% offline probability. When each round witnesses one random client offline, the performances of two methods are below 45% while the improvement brought by our approach is only 0.4%. That reveals the performance boosts of PrivacyFace can be influenced by the network conditions.

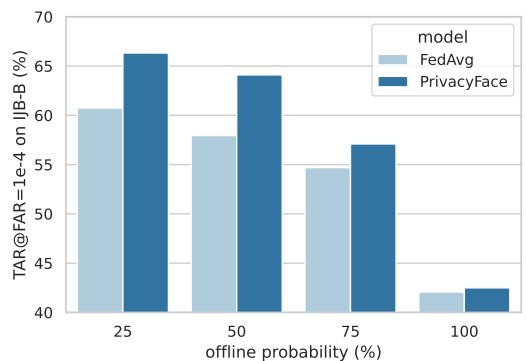

Figure 9: Effects of client disconnections during training.

## A.3 DIFFERENCES WITH DIFFERENTIAL PRIVATE FEDERATED LEARNING

In PrivacyFace, we utilize the differential privacy to improve the performance of conventional federated learning in face recognition. Our work basically distills differentially private information from clients to achieve that purpose. We notice that there are plenty of recent works on differential private federated learning (DP-FL) methods (Geyer et al., 2017; McMahan et al., 2017b; Truex et al., 2019) and would like to emphasize the differences:

1. PrivacyFace and current DP-FL methods aim to solve different privacy leakages. As different data points lead to different model updates, DP-FL methods essentially add noise to model update to prevent data from being inspected. Recall the model update is essentially the average of gradients, which has a small $l_2$-sensitivity (definition 2) thanks to the tremendous number of data points. Then the protection is relatively easy with the low privacy cost (propotional to the $l_2$-sensitivity). In contrast, PrivacyFace aims to protects class centers in face recognition which are directly linked to individual privacy. The problem is much harder as these class centers are not naturally averaged like model updates and therefore leads to large privacy cost.

2. The targets to protect is different. Current DP-FL methods protect the data points in each local dataset. In contrast, PrivacyFace protects some of the model parameters (*i.e.*, the classifier weights), which is the first work in federated learning to our best knowledge.

3. Current DP-FL methods theoretically decrease performances of federated learning as they use noisy updates. However, PrivacyFace distill differentially private information from each client. The extra information can broadcasted to improve the performances of federated learning face recognition.

4. We can directly combine DP-FL method and PrivacyFace, which lead to a more secure framework protecting both data points as well as class centers together.

