# OpenReview forum: "Improving Federated Learning Face Recognition via Privacy-Agnostic Clusters"
_ICLR.cc/2022/Conference — ICLR 2022 Spotlight_

### Official Review · Reviewer_7nj6 · 2021-10-28

**Correctness:** 4
**Technical Novelty And Significance:** 4
**Empirical Novelty And Significance:** 4
**Recommendation:** 8
**Confidence:** 5

**Main Review:**

+ The proposed method can well address the issue of privacy leakage.
+ The face recognition performance benefits from the supervision combination of local data and online cluster.
- We doubt whether the gaussian perturbation is able to cover the privacy of facial ID. How to balance the privacy preserving and the training effect through tuning the extent of the gaussian perturbation? This is not clear in the paper.
- The urgency of the problem itself is not solid to me. Even though the value of $W^c$ is leaked, the ID information cannot be easily recovered without the remaining part of the neural network.


**Summary Of The Paper:**

The aim of this paper is to tackle the privacy leakage issue when using federated learning technique to train deep face recognition models. The naïve practice, broadcasting the last fully connected layer, results in the leakage of ID features. But if we cut off the broadcast of $W^c$, it will cause the issue of overlapping IDs among the clients, which harms the training of face recognition model. Therefore, the authors propose their method, DPLC, to tackle this problem. The idea is to conduct online clustering, gaussian perturbation, combined supervision of cluster and local data.

**Summary Of The Review:**

Based on the primary opinion above, I will temporarily give the ranking of “marginally below the acceptance threshold”. The final rating will depend on the authors’ feedback.

---

> ### Author Response · Authors · 2021-11-16
> **Response to Reviewer 7nj6**
>
> We thank the reviewer for the time reviewing our work and for the insightful comments. Please find our explanations about how Gaussian perturbations can cover the privacy in *General Author Response to Privacy-related Comments*. In a nutshell, the generated cluster centers are not that sensitive as they are basically group information. The Gaussian perturbation further creates $(\epsilon, \delta)$-differentially private outputs, which have rigorous theoretical guarantees of privacy. Below please find our responses to other comments:
>
> - **The balance between privacy and utility.** The trade-off is discussed in Fig. 5(b). The extent of Gaussian noise is controlled by the privacy cost $\epsilon$ as shown in Theorem 2. A large privacy cost $\epsilon$ leads to little noise, which consequently results in better performances.
> - **The urgency of the problem.** We agree that it's not easy to recover privacy without the backbone network. However, for federated learning face recognition, all local backbone networks will be uploaded to the server. In addition, backbone networks in all clients will be similar when the training converges. Recall that the goal of federated learning is to collaboratively train a model without privacy leakage for all involved clients. That means any other client or the server shouldn't be able to learn the local data of a client. Therefore, the exposure of $W^c$ will lead to privacy leakage.

---

> > ### Comment · Reviewer_7nj6 · 2021-11-20
> > **I raise the rating to Accept**
> >
> > Thank you for your response which addresses our concerns well. We change the rating to Accept.

---

> > > ### Author Response · Authors · 2021-11-20
> > > **Thank you for raising the rating**
> > >
> > > We are glad that concerns are addressed and appreciate the reviewer for raising the score.

---

### Official Review · Reviewer_ytHU · 2021-11-03

**Correctness:** 3
**Technical Novelty And Significance:** 3
**Empirical Novelty And Significance:** 3
**Recommendation:** 8
**Confidence:** 4

**Main Review:**

This work proposes a novel framework to train a face recognition network for the federated learning setting.
The points of the strength for the work are listed as follows:

1. the proposed differentially private local clustering mechanism with solid theoretical guarantees allows to securelly share the information of class centers of a local client to other clients.
2. In addition, the consensus-aware recognition loss encourages the global consensuses for the learned face embeddings among clients and significantly improves the performance as compared with the baselines.

The points of the weakness:

1. some parameters for the experiments are missing. I could not find the number of clients participating in the training. In addition, is the data distributed in iid or non-iid way. How many clients (the percentage) do participate in the training for each communication round? The details of these setting and other experimental results and discussion are totally missing from the paper.
2. It seems that the training requires to start with a pretrained face recognition model using some large-scale dataset. However, for the current face recognition models, most of them can be trained from scratch even using the CASIA-WebFace instead of using the MS1M dataset. The bad initial class centers could still lead to the issue as shown in Fig. 1(a) where the embedding space could still overlap. The studies of different initial pretrained models \phi_{0}'s are missing to show if the proposed method is senstivie to different initial models.

**Summary Of The Paper:**

This work proposes a novel framework to train a face recognition network for the federated learning setting. The proposed method leverages differentially private local clustering mechanism to allows to securelly share the information of class centers of a local client to other clients.
 In addition, the consensus-aware recognition loss encourages the global consensuses for the learned face embeddings among clients and significantly improves the performance as compared with the plain federated learning baselines.

**Summary Of The Review:**

I think the idea is novel and there are many solid theoretical proofs for differentially private guarantee to secure the information exchange among clients for model training. However, some important parts for federated learning or face recognition are still not well explained in the current paper. Since the federated learning is to train the model distributedly, some clients may drop for certain round of communication due to network issues. However, from algorithm 2, the experimental results are mainly done in the perfect setting that each round, all clients participate in the training and share their clusters to others through the server. It can be imagined that if some clusters do not participate in the training for a certain round, the consensus-aware recognition may fail and still result in the same situation as shown in Fig. 1(a). Thus, It is worth a study how it  would influence the final performance.

Does the author try different initial models to start training? Does the proposed framework require a strong face model to begin with (trained with a larger dataset and finetuned onto a smaller or similiar-size dataset)? Is it possible to start from scratch or a pretrained model with smaller dataset (CASIA-WebFace) and then to finetune onto a larger dataset (MS1M)?

Can the model scale to train with MS1M or deepGlinit or recent released WebFace260M which contains much more subjects (class centers) than the CASIA-WebFace (10575) and BUPTBalancedFace?

I will adjust my rating according to the feedback from the authors.

---

> ### Author Response · Authors · 2021-11-16
> **Response to Reviewer ytHU**
>
> We thank the reviewer for the constructive suggestions and feedback and feel honored for being approbated on the theoretical parts of this work. The reviewer's comments help us a lot to improve our work. We provide discussions on the reviewer's concerns as follows:
>
> - **Some parameters.** We have updated the training configurations in the revised version. In the original draft, the number of clients is given two lines below Fig. 4 as "We ergo assign each of the sub-datasets to one of four clients". The data is distributed in a non-iid way as there is a prior probability shift (datasets are demographically partitioned) in BUPT-BalancedFace. Also, we use the perfect setting where all clients participate in the training for all communication rounds.
> - **Effects of Initial models.** Thanks for the valuable advice. We do not require a strong initial model as the $\phi_0$ Is trained on CASIA-WebFace which is a small dataset. We add Sec. A.2.6 in the appendix which presents results with four different initial models. With a weak initial model $\phi_2$, PrivacyFace can still achieve 3.43% boosts on TAR@FAR=1e-4 on IJB-C. The improvement is not as significant as with $\phi_0$, which may be because the training loss is overwhelmed by poor local features and the global consensus is of secondary importance. If using a strong model $\phi_3$ where the global consensus is already achieved, the training also shifts to achieve better local features. In this case, the highest improvements on RFW are observed.
> - **Effects of client disconnections.** Thanks for the valuable advice. Sec. A.2.7 is added in the appendix to study this issue. Generally, the performance boosts brought by our framework are less significant when more disconnections occur during training.
> - **Scalability.** We believe that PrivacyFace can work well in large-scale settings because the global consensus becomes more important with more clients involved. However, we will probably leave it for future work for the following reasons:
>   1. Currently, BUPT-Balancedface and RFW are the most suitable face datasets for the federated setting. For detection, several real-world datasets have been collected in various locations [1,2] for the federated learning task.  For person reID, current benchmarks (*e.g.*, MSMT17, DukeMTMC, Market-1501) are collected in different scenes and naturally fit the federated setting [3, 4].  However, the existing large-scale face datasets (*e.g.*, Glint360K, WebFace260M) are all from the websites and cannot be directly used. Many efforts (*e.g.*, add extra labels and split the datasets) are required to find proper protocols for federated learning.
>   2. Training on these datasets requires substantial computational cost, which is not practical considering the limited time and GPU resources we have.
>
> [1] FedVision: An Online Visual Object Detection Platform Powered by Federated Learning
>
> [2] Real-World Image Datasets for Federated Learning
>
> [3] Performance Optimization for Federated Person Re-identification via Benchmark Analysis
>
> [4] Decentralised Learning from Independent Multi-Domain Labels for Person Re-Identification

---

> > ### Comment · Reviewer_ytHU · 2021-11-21
> > **I change my rating.**
> >
> > The authors have answered most of my concerns and make changes. I have changed my rating to accept.

---

> > > ### Author Response · Authors · 2021-11-21
> > > **Thank you for raising the rating**
> > >
> > > We are glad that most concerns are addressed and appreciate the reviewer for raising the rating.

---

### Official Review · Reviewer_ZPue · 2021-11-04

**Correctness:** 2
**Technical Novelty And Significance:** 2
**Empirical Novelty And Significance:** 2
**Recommendation:** 5
**Confidence:** 5

**Main Review:**

1) First and foremost, the motivation and threat model of the proposed scheme is not clear. For example, what is the exact privacy goal of a participant in federated face recognition? Is it to prevent other participants from learning about the faces/identities in the local dataset? If yes, how does differential privacy help in achieving this goal? Given the updated feature embedding model from the server AND the 'noisy' cluster centers of the weight vectors, isn't it fairly straight forward to reverse engineer the face images that correspond to each cluster?

2) Secondly, the proposed scheme is likely to work if all the face images in the local dataset are somewhat similar and form a compact cluster. Moreover, the face images in the other local datasets must not have large overlap with the local face images in the feature space. In the absence of these constraints, it may be hard to reach global consensus when the participants have mixed datasets (a more realistic setting). This may be the reason the experiments are conducted based on each participant having samples only from a single race.

3) The paper appears to analyze the differential privacy guarantees of revealing the cluster centers and the model parameter updates of the feature embedding separately. However, these two strands of information may be used in conjunction to develop more sophisticated attacks.

4) The experimental validation is weak. The proposed approach has been compared only against a centrally trained model and the proposed model without differential privacy. It has not been benchmarked against recent works on federated face recognition such as:

D. Aggarwal, J. Zhou and A. K. Jain, "FedFace: Collaborative Learning of Face Recognition Model", International Joint Conference on Biometrics (IJCB) 2021

**Summary Of The Paper:**

The main contribution of the paper is a slight modification of the standard federated learning (FL) framework for the purpose of improving face recognition performance. Since face recognition involves learning a deep neural network for feature embedding and the weight vectors for mapping the feature embedding to an identity, the standard FL framework (e.g. FedAvg) that updates only the feature embedding network does not suffice. Hence, the paper proposes to cluster the weight vectors of each participant and transmit the cluster centers (albeit with Gaussian noise to ensure differential privacy) to the other participants. Once the cluster centers of other participants are known, the local weight vectors can be learned to avoid overlap with those received clusters.

**Summary Of The Review:**

The motivation and threat model of the proposed work are not clear. The proposed approach appears to have some practical limitations, which have not been considered. The privacy analysis is not very robust and the experimental validation is also weak.

During the rebuttal process, concerns 2 and 4 have been addressed to a great extent. Concern 1 still remains a theoretical possibility, but given that I do not have strong evidence to disprove the claims in the paper, it is fair to give the benefit of doubt to the authors. Concern 3 is still a very valid concern, but has been dismissed lightly in the rebuttal. Overall, I would like to upgrade my rating by one level based on all the discussion.

---

> ### Author Response · Authors · 2021-11-16
> **Response to Reviewer ZPue**
>
> We thank the reviewer for the time reviewing our work and for the insightful comments. The concerns of the reviewer lie in the privacy issue and the experiments. Please find our response to the privacy issue in *General Author Response to Privacy-related Comments*. We also explain the baseline issue in *General Author Response to Comments on Baselines*.  Below are our responses to other comments:
>
> - **Motivation.** We believe our motivation is clear enough in the draft. In general, our target is to resolve the paradox in the abstract: "broadcasting class centers among clients is crucial for recognition performances but leads to privacy leakage". Federated learning is proposed to convince clients to collaboratively contribute to a general model without exposing their local data. Therefore, our answer is yes to the question, "is it to prevent other participants from learning about the faces/identities in the local dataset".
>
> - **Threat model.** With a differential privacy mechanism, we no longer need attack modeling [1] because DP theoretically is resistant to any threat model. Sec. A.2.4 of the appendix presents two threat models (K-nearest neighbor attack and reversion attack) as examples.
>
> - **It's straight forward to reverse engineer images from cluster centers.** We agree that one can directly reconstruct an image from the cluster center, also shown in Fig. 8 of the appendix. Theoretically, noisy cluster centers are privacy-agnostic because differential privacy prevents all attacks (including reverse engineering) from inspecting privacy. Experimentally, Fig. 8 in the appendix has shown that the reconstructed images reveal little information.
>
> - **Local dataset are somewhat similar and form a compact cluster.** PrivacyFace does not require such a strong assumption. As shown in Fig. 4(c),  we can query 3 clusters from the sub-dataset with only 3000 out of 7000 classes involved. The remaining classes can be dissimilar to those in the clusters. That also means PrivacyFace can work if large class overlaps exist among local datasets, as long as they are not included in the queried clusters. Moreover, if datasets from two clients are highly similar, the conflicts can be easily circumvented by ignoring cluster centers from each other when using the consensus-aware loss.
> Generally, our assumption is that some classes of datasets can be clustered with local properties. We admit this assumption can not fit all federated settings. However, it's still applicable in many real-world scenarios because faces from one client can be captured in similar domains (*e.g.*, from similar cameras, lighting conditions and physical locations) or from people with similar characteristics (*e.g.*, race, gender, age, expression). Those local similarities are regarded as group information, which is utilized to improve the model performances in our work.
>
>
>   [1] https://desfontain.es/privacy/differential-privacy-awesomeness.html

---

### Official Review · Reviewer_VU44 · 2021-11-06

**Correctness:** 3
**Technical Novelty And Significance:** 4
**Empirical Novelty And Significance:** 3
**Recommendation:** 8
**Confidence:** 3

**Main Review:**

1. I think this is an impactful work. It aims to solve a practical and challenging task: FL for face recognition (FR). To my knowledge, in industry, the face recognition companies have not (at least, not widely) deployed the edge/distributed learning for face recognition products. Thus, this is a poineering work. The authors seem understand both FL and FR in depth, the proposed solution can meet the requirements of differetiable priacy and improve the face recognition performance in the edge/distributed learning scenarios.
2. I am from the filed of FR rather than FL. I did not check very carefully the proof and claims of the solutions meet the requirements of differential priacy. I just quickly went through those proof and claims, looked sound for me.
3. I have very minor concerns for the experiments. (1) The authors do not compare with any existing methods. Then the proposed method looks like a black box in terms of performance. (2) The performance still drops greatly comopared with training using all the client data. (3) More analysis experiments are needed to help people to understand every component of the proposed method.
4. I am not the expert on FL, then I have some questions on that.  (1) In introduction section, it is claimed 'FL decentralizes the training process by combining local models fine-tuned on each client’s private data and thus hinders privacy breaches.' Does it mean, traditinoally, one client cannot add other clients' face information at all for improving one client's training  due to the privacy problem? Traditionally, can the server collect the face information from all the clients and use them for training? If no for both questions, does it mean this work is the first to ask the client and server to use others' face information to improve the training? (2) 'That prevents the FL approach from broadcasting the whole model amongclients and the central server, and consequently leads to conflicts in the aggregation of local updatesin the overlapped regions. ' What do you mean the ovelapped regions?  (3) The authors mentioned privacy cost and the proposed method is efficient. Can you define what is the privacy cost? Which procedure does this cost happen?

**Summary Of The Paper:**

This paper proposes a FL strategy for face recognition. Despite the wide investigation of FL, very littler research discusses the use of FL for face recognition.  The proposed of PrivacyFace can: (1) distill sanitized clusters from local class centers using Differentially Private Local Clustering (DPLC) . (2) use a consensus-aware recognition loss to  encourage global consensuses among clients, leading to a more discriminative feature learning.


**Summary Of The Review:**

As aformentioned, I think this is an impactful work. The authors understand both FR and FL in depth and proposed a convincing solution. I think this work is inspiring for both industry and academia.

---

> ### Author Response · Authors · 2021-11-16
> **Response to Reviewer VU44**
>
> We thank the reviewer for the positive assessment of the paper, especially the remark "inspiring for both industry and academia". We are mainly from the FR field as well and believe that privacy-guaranteed approaches are urgently demanded to restore public confidence in the field. We expect PrivacyFace to be one of such approaches and to assist future works in FR. Please find our baseline and privacy cost responses in *General Author Response to Comments on Baselines* and *General Author Response to Privacy-related Comments*. Responses to other concerns are as follows:
>
> - **Performance degradation.**  We agree that PrivacyFace does not perfectly solve the privacy-utility paradox, and hope our work can inspire future works for better solutions. Experimentally, we would emphasize an advantage of our method is that it significantly improves the plain FL with very little overhead introduced. The cost analysis can be found in the last part of our experiment as well as the newly added Sec. A.2.5 of the appendix.
> - **More analysis experiments on components of the method.** We are very willing to add analysis experiments to help readers understand our work if there is any suggestion. Currently, we examine how hyper-parameters (privacy cost $\epsilon$, occupation margin $\rho$ and query number $Q$) affect utility of $\hat p$ and recognition performances as shown in Fig. 3, 4,5. Sec. A 2.1-2.4 of appendix present some other analyses.
> - **Should face information be exchanged?** We would recommend not exchanging private face information among clients or between a client and the server. FL is designed to convince clients that they can collaboratively train a general model without privacy leakage. Distributing private face information breaches that goal. In PrivacyFace, our solution is to exchange insensitive group information to improve model performances.
> - **First to improve the training by face info?** Our answer is no and yes. The reason for no is that FedFace [1] also attempts to improve the training by broadcasting face information. Specifically, it uploads all local class centers to the server and separates them in the server. However, that operation is unsafe as the server can potentially know all involved identities in each client, which can be verified by attacking examples in Sec. A.2.4 of the appendix. For our work, we would say it is the first to use **secure** face information to improve the training.
> - **Overlapped regions.** Thanks for pointing it out. In the global distribution of Fig. 1(a), class centers from client 1 and C have an overlapped area described as "overlapped regions" in the original draft. This seems a little confusing and we have revised that sentence.
>
> [1] D. Aggarwal, J. Zhou and A. K. Jain, "FedFace: Collaborative Learning of Face Recognition Model", International Joint Conference on Biometrics (IJCB) 2021

---

### Author Response · Authors · 2021-11-16
**General Author Response to Privacy-related Comments**

We thank the reviewers for their time reviewing our work and for the pertinent and constructive comments. We found that there are several comments about privacy and would like to clarify them together. Here are these questions:

1. Noisy cluster centers can compromise privacy by reverse engineering (**Reviewer ZPue**).
2. In the future, they may be more sophisticated attacks to spy out privacy from the sanitized cluster centers and the backbone (**Reviewer ZPue**).
3. The Gaussian perturbation may not be able to cover the privacy of facial ID (**Reviewer 7nj6**).
4. Meaning of the privacy cost (**Reviewer VU44**).

We'd like to start with the definition of privacy in differential privacy (DP). The privacy goal of DP is to simultaneously protect every individual row while permitting statistical analysis of the database as a whole (page 11 in Dwork *et al.* [1]). Therefore, group information (page 23 in Dwork *et al.* [1]) is not private. Taking the randomized response in the tutorial [2] as an example, whether a participant is an illegal drug user is private, while the number of users is not. Therefore, for **question 1**, reverse engineering images from the cluster centers will only compromise group information, which is not directly related to privacy.

However, privacy can be spied out from plain group information. Suppose there are many bankrupt people and one billionaire, and we report mean asset of 10 randomly selected people. If the mean value is large, we will know the billionaire is sampled, and vice versa. To alleviate the risk, the DP mechanism generates "basically the same" outputs if you change the data of one individual [2] (in this example, the change means whether the billionaire is selected). Definition 1 in our paper shows the formal definition, which states neighboring datasets will generate every possible output with similar probability (*i.e.*, $P[M(X)\in T] \leq e^\epsilon P[M(X'\in T) + \delta$). By this means, the attacker can no longer inspect individual privacy.

In our work, we propose a DPLC mechanism to generate noisy cluster centers. The mechanism theoretically achieves $(\epsilon, \delta)$-DP  with Gaussian noise with enough variance, as shown in Theorem 2. Therefore, for **question 1** and **question 3**, we believe that the noisy cluster centers are highly secure and won't compromise privacy.

For **question 2**, we would say that the case is less likely to happen as differentially private outputs are statistically consistent w/ or wo/ any individual in the dataset. The tutorial [3] states that no attack modeling is needed to be worried as DP protects any kind of individual information, irrespective of the attacker’s prior knowledge, information source and other holds [1]. Even if such sophisticated attacks exist in the future, we cannot deny the security of our method at the moment. There is a similar case in cryptography. It's well-known that most crypto-systems (besides those based on lattice cryptography) will become unsafe if quantum computers are designed. However, we can still trust these crypto-systems until then.

For **question 4**, we like the intuitive explanation of the privacy parameter $\epsilon$ (*a.k.a.*, the privacy cost) here [4]. We should let $\epsilon=0, \delta = 0$ to make all datasets have the same outputs in an ideal situation. However, the discrepancy is inevitable in most cases, and that discrepancy is quantized by privacy cost (Definition 1 in our paper). High privacy cost leads to inconsistent outputs statistically, which attackers can utilize. Besides the quantization of privacy, another good property of DP is the composition rule (Definition 4 in the paper). We use that property in the cost analysis (the last part of our experiments in the main text) to say that privacy cost increases with communication rounds. An intuitive explanation is that if there are infinite DP queries from a dataset, the Gaussian noise can be canceled out by averaging the outputs, which will also expose the plain output and consequently lead to privacy risk. That's also why we choose to fine-tune the model instead of training from scratch, as the former approach reduces the privacy cost with fewer communication rounds.

We tried our best to simplify the understandings of DP. However, there may still exist vague statements in view of the abstruseness of DP. Please do not hesitate to post further comments or questions.

[1] Dwork, Cynthia, and Aaron Roth. "The algorithmic foundations of differential privacy." *Found. Trends Theor. Comput. Sci.* 9.3-4 (2014): 211-407.

[2] https://desfontain.es/privacy/differential-privacy-in-more-detail.html

[3] https://desfontain.es/privacy/differential-privacy-awesomeness.html

[4] https://crypto.stackexchange.com/questions/44739/intuitive-explanation-of-the-varepsilon-parameter-in-differential-privacy

---

> ### Comment · Reviewer_ZPue · 2021-11-18
> **Response to Privacy-related Comments**
>
> While the authors have provided a long explanation on differential privacy, it has been often shown in the literature that a differential privacy guarantee is meaningless if the privacy budget $\epsilon$ is sufficiently large. The fundamental question is whether a value of $\epsilon = 1$ used in this paper sufficient to preserve privacy in this application. Moreover, all the cited intuitive examples about differential privacy (e.g., number of illegal drug users, bankrupt vs. billionaire, etc.) make perfect sense in a univariate world. In this paper, the goal of the adversary is to recover/reconstruct face images, which is usually high-dimensional with a strong structural correlations among the attributes. The adversary can also build a reasonable prior model from public data. In such a scenario, noisy cluster centers may be good enough to sufficiently constrain the search space and find possible face images that form the cluster.

---

> > ### Author Response · Authors · 2021-11-18
> > **Further Response to Reviewer ZPue**
> >
> > We thank the reviewer for the quick response. Below are our responses to new comments:
> >
> > - **Is $\epsilon=1$ sufficient?**  We'd like to list privacy costs for several recent works: 0.5-1.5 for Luo *et al.*,  2,93 for Private-kNN[2], 0.83-5 for PATE[3], 2-8 for Papernot *et al.* [4]. Therefore, $\epsilon=1$ is at least a competitive value to current literatures. Moreover, whether $\epsilon=1$ is enough is decided by the real-world situations instead of by the algorithm itself. Some applications have high budgets while others don't, and we need to adjust the $\epsilon$ in accord with specific conditions. .
> > - **Only easy examples like drug users are provided.** In our previous response, we present intuitive examples just in case that some reviewers were unfamiliar with DP. That does not mean DP cannot be applied to the cases where "high-dimensional with a strong structural correlations among the attributes", as described by the reviewer. The listed works [1,2,3,4] have successfully applied DP for training deep networks and conducted experiments on large classification, face and ReID benchmarks (*e.g.*,  CIFAR-10, CS-FSL, MINIST, SVHN, CelebA, Market1501 and UCI Adult).
> > - **The adversary can find possible face images.**  We'd like to emphasize that DP is not proposed to stop the adversary from inferring "possible" data points. The adversary can have as many guesses as possible. The concept of "privacy" in DP is that the adversary is unknown about the correctness of a guess, as differentially private outputs are "basically" the same no matter whether a guess is in the dataset.
> >
> > [1] Luo, Zelun, et al. "Scalable Differential Privacy With Sparse Network Finetuning. CVPR2021.
> >
> > [2] Zhu, Yuqing, et al. "Private-knn: Practical differential privacy for computer vision." CVPR2020.
> >
> > [3] Papernot, Nicolas, et al. "Scalable Private Learning with PATE." ICLR2018.
> >
> > [4] Papernot, Nicolas, et al. "Semi-supervised knowledge transfer for deep learning from private training data." ICLR2016.

---

> > > ### Comment · Reviewer_ZPue · 2021-11-19
> > > **Further response to privacy discussion**
> > >
> > > 1) In my humble opinion, privacy budgets across different applications/papers are not comparable. For example, a specific $\epsilon$ may be good enough for an application, where the adversary has no (or little) prior knowledge about the characteristics of training samples that he/she is trying to recover/reconstruct. The same $\epsilon$ may not be sufficient for a different application like face recognition, where the adversary has very good prior knowledge based on public datasets. So, the question of whether $\epsilon = 1$ is sufficient to preserve privacy of face images is a valid one.
> > >
> > >  2) While Figure 8 in the appendix does show that the so-called "reversion" attack is not so successful, the main concern is about more sophisticated attacks such as the one presented in:
> > >
> > > Y. Zhang, R. Jia, H. Pei, W. Wang, B. Li and D. Song, "The Secret Revealer: Generative Model-Inversion Attacks Against Deep Neural Networks," 2020 IEEE/CVF Conference on Computer Vision and Pattern Recognition (CVPR), 2020, pp. 250-258.
> > >
> > > While the above reference assumes white-box access to the model, the proposed framework gives "almost" white-box access (the entire feature extractor is known along with a "noisy" version of the classification layer). According to the above reference, even stricter privacy budgets such as $\epsilon = 0.1$ may not be sufficient to prevent reconstruction attacks.
> > >
> > > 3) It is true that the concept of "privacy" in DP is that the adversary does not know about the correctness of a guess. This is why my original question was about the motivation/goal of the adversary. If we say that the goal of the adversary is to exactly reconstruct the original face images available in the local dataset of another participant with 100% certainty, then the proposed method can be considered as secure because the adversary has no way of validating his guesses (neither exact reconstruction nor 100% certainty is possible). On the other hand, if the goal of the adversary is to infer the identities available in the local dataset of another participant with a fairly good degree of accuracy, it is quite plausible to achieve this goal. Note that face images are not secrets - once the adversary has a good approximation of the victim's face, it is often easy to identify the victim.
> > >
> > > 4) Finally, the whole discussion has been centered around privacy leakage associated with sharing of cluster centers (parameters of the classification layer). It would also be beneficial to know the privacy leakage associated with sharing of updates required to learn the common feature extractor. Several recent studies on gradient leakage have show that gradients of layers closer to the input and at early stages of training can severely compromise privacy.

---

> > > > ### Author Response · Authors · 2021-11-19
> > > > **Further Authors' Response about Privacy to Reviewer ZPue**
> > > >
> > > > Thanks for the response and here are our responses.
> > > >
> > > > - **Whether privacy cost $\epsilon=1$ is sufficient.**
> > > >   1. Again, DP protects any kind of individual information, irrespective of the attacker’s prior knowledge, information source, and other holds [1]. The privacy budget won't be affected by whether the adversary "has very good prior knowledge based on the public dataset".
> > > >   2. We confirmed with the authors of Zhang *et al.* [2] that their attacking model is based on the feature extractor and the classifier parameters together. Those classifier parameters contain each individual's class center, from where the individual privacy can be obtained. Therefore, the assumption is that they can get access to an individual's class center, corrupted or blurred images. With such individual information, the attacker surely can attempt to spy out privacy. However,  we only generate a cluster center (not a "noisy" version of the classifier as understood by the reviewer), which is not linked to any individual. That protects the privacy from the source.
> > > >   3. We also confirmed with the authors of Zhang *et al.* [2] their statement about "$\epsilon=0.1$ may not be sufficient" is not to deny the efficacy of DP. Their point is classifier parameters of the trained differentially private models can be utilized by attackers, as the DP they used is to protect the dataset during training instead of protecting the classifier. Moreover, the privacy risk from the classifier in Zhang *et al.* [2] further verifies our motivation of protecting the parameters.
> > > >   4. Therefore, we disagree with the reviewer's statement "it is quite plausible infer the identities with a fairly good degree of accuracy". The attackers such as Zhang *et al.* [2] can work only if individual information is available, while our work only exposes differentially private group information. The attacker cannot know whether an individual is in the dataset from the group information, which is the meaning of not knowing the correctness of the guess.
> > > >   5. The statement of Zhang *et al.* [2] "$\epsilon=0.1$ may be not sufficient" is based on their experiments on MNIST. If the reviewer denies the comparisons of privacy cost across tasks in our previous response, how can "$\epsilon=0.1$ may be not sufficient" be valid in face recognition?
> > > >   6. We believe which privacy cost is good enough should be decided by real-world situations instead of the algorithm itself. You can choose to use $\epsilon=0.1$, which can also offer over 4% improvements as shown in Fig. 5(b). Our contribution is the proposed new framework instead of the $\epsilon$ itself.
> > > > - **Privacy leakage from model updates.** That issue can be addressed by DP-FL algorithms and is irrelevant to our work. Our work is to improve the model performance by safely sharing group information. We have talked about the differences in Sec. A.3 of the appendix.  Moreover, one can directly combine DP-FL method and PrivacyFace to protect both training datasets and classifier parameters.
> > > >
> > > > [1] Dwork, Cynthia, and Aaron Roth. "The algorithmic foundations of differential privacy." *Found. Trends Theor. Comput. Sci.* 9.3-4 (2014): 211-407.
> > > >
> > > > [2] Y. Zhang, R. Jia, H. Pei, W. Wang, B. Li and D. Song, "The Secret Revealer: Generative Model-Inversion Attacks Against Deep Neural Networks," 2020 IEEE/CVF Conference on Computer Vision and Pattern Recognition (CVPR), 2020, pp. 250-258.

---

> > > > > ### Comment · Reviewer_ZPue · 2021-11-20
> > > > > **Final Comment**
> > > > >
> > > > > Thanks for the response. We can keep discussing these issues for ever. But here are my final thoughts.
> > > > >
> > > > > 1) When FL was introduced, it was argued that gradients/model updates do not compromise privacy. Later, it was argued that aggregated/averaged gradients do not compromise privacy. Today, we are faced with a situation where aggregated gradients with differentially private noise added to them may not be able to achieve a good privacy-utility tradeoff.
> > > > >
> > > > > The scenario in this paper is analogous. We know that class centers leak information and even cluster centers leak a significant amount of information. So, in this paper, cluster centers with differentially private noise has been suggested. I believe it is only a matter of time that someone comes up with sophisticated reconstruction attack that renders the differential privacy guarantees meaningless. Anyways, this is a theoretical argument, and I understand that it does not carry much weight unless supplemented with practical evidence.
> > > > >
> > > > > 2)  But the other claim the privacy leakage from model updates can be dealt with separately and is irrelevant to this work is quite concerning. Also, this dichotomy between protecting training datasets and classifier parameters is misleading. The only reason someone is interested in protecting the classifier parameters is because they leak information about the training datasets (face images used to train the classifier parameters). Since the eventual goal is to protect the faces used in training, all the sources that leak information must be considered together. Otherwise, it may be possible to easily mount a gradient leakage attack (Geiping et al., NeurIPS 2020) on the feature extractor part with the information available from the "noisy" cluster centers as additional constraints.

---

> > > > > > ### Author Response · Authors · 2021-11-20
> > > > > > **We appreciate the discussions with the reviewer.**
> > > > > >
> > > > > > Thanks for the comment and we appreciate the discussions with the reviewer. We like the examples on FL where attacking and defending models stand up against each other, which prompts the alternative developments of both sides. During the process, many insightful works are proposed. However, if we deny defending models with potential future unsafety, this field would stagnat.
> > > > > >
> > > > > > Similarly, even though that DP and our PrivacyFace may be unsafe in the future, its current security still contributes to the community with potential applications and catalyzing more sophisticated attacking/defending models.

---

### Author Response · Authors · 2021-11-16
**General Author Response to Comments on Baselines**

We thank reviewers for their time and efforts. We'd like to respond to concerns about baselines from **Reviewer VU44** and **Reviewer ZPue** together. In our experiments, we only choose the conventional FL approach as the baseline. The reason is that we cannot find another approach that can be applied in our setting with privacy guarantees. FedFace seems to be a related method (also pointed by **Reviewer ZPue**), which uploads all local class centers to the server and then forces all gathered class centers to be discriminative. However, the server can then know all involved identities in each local dataset by some attacking methods (*e.g.*, K-Nearest Neighbor Attack or Reversion Attack as shown in Sec. A.2.4 of the appendix). Recall that federated learning, which enables collaborative training among clients without sharing data, is proposed for privacy purposes. If with such privacy risk, clients won't be convinced to participate in the federated training.

We also cannot find proper baselines from other fields, which may be caused by the peculiarity of face recognition where model parameters are related to privacy. Therefore, we only use the conventional federated learning in the initial version. However, we agree that it would be interesting compare FedFace and our method, and conduct the experiments in the updated draft (Sec. A. 2.5 of the appendix). The main results are summarized in the following table, where PrivacyFace surpasses FedFace in recognition performances, privacy cost and additional costs (including communication cost, computational cost and extra GPU memory used). Note that we only sample 1000 out of 7000 class centers in each client for the spreadout regularizer in FedFace. If using all class centers, the additional GPU memory is around 11,962M, which is out of the capacity of our machine.

| Method      | Performance on IJB-C | Privacy Cost | +Communication Cost | +Computational Cost | +GPU memory |
| ----------- | -------------------- | ------------ | ------------------- | ------------------- | ----------- |
| FedAvg      | 60.98                | 0            | 0                   | 0                   | 0           |
| FedFace [1] | 63.25                | $+\infty$    | 14,000 Vectors      | High                | 1525M       |
| PrivacyFace | 71.24                | 1            | 5 Vectors           | Low                 | <1M         |

[1] D. Aggarwal, J. Zhou and A. K. Jain, "FedFace: Collaborative Learning of Face Recognition Model", International Joint Conference on Biometrics (IJCB) 2021

---

### Author Response · Authors · 2021-11-16
**Changes to the draft**

We thank reviewers for their thorough reviews and valuable comments!  We have revised our paper (changes are highlighted in blue) by taking into account some of their suggestions. The major updates are summarized below.

- Add experiments on FedFace in Sec. A.2.5 of the appendix (suggested by Reviewer VU44 and ZPue).
- Add the ablation study on the initial models in Sec. A.2.6 of the appendix (suggested by Reviewer ytHU).
- Add experiments on how client disconnections influence performances (suggested by Reviewer ytHU).
- Add remarks on experimental settings (suggested by Reviewer ytHU).
- Remove the "overlapped region" in the introduction (suggested by Reviewer VU44).

Please do not hesitate to post further comments or questions.

---

### Decision · Program_Chairs · 2022-01-20

**Decision:**

Accept (Spotlight)

**Comment:**

This paper received 4 quality reviews. The rebuttal and discussions were effective. All reviewers raised their ratings after the rebuttal. It finally received 3 ratings of 8, and 1 rating of 5. The AC concurs with the contributions made by this work and recommend acceptance.